# CONTROLLABLE BLUR DATA AUGMENTATION USING 3D-AWARE MOTION ESTIMATION

**Insoo Kim[1,2]  Hana Lee[1]  Hyong-Euk Lee[1]  Jinwoo Shin[2]***
[1]SAIT, Samsung Electronics [2]KAIST

## ABSTRACT

Existing realistic blur datasets provide insufficient variety in scenes and blur patterns to be trained, while expanding data diversity demands considerable time and effort due to complex dual-camera systems. To address the challenge, data augmentation can be an effective way to artificially increase data diversity. However, existing methods on this line are typically designed to estimate motions from a 2D perspective, e.g., estimating 2D non-uniform kernels disregarding 3D aspects of blur modeling, which leads to unrealistic motion patterns due to the fact that camera and object motions inherently arise in 3D space. In this paper, we propose a 3D-aware blur synthesizer capable of generating diverse and realistic blur images for blur data augmentation. Specifically, we estimate 3D camera positions within the motion blur interval, generate the corresponding scene images, and aggregate them to synthesize a realistic blur image. Since the 3D camera positions projected onto the 2D image plane inherently lie in 2D space, we can represent the 3D transformation as a combination of 2D transformation and projected 3D residual component. This allows for 3D transformation without requiring explicit depth measurements, as the 3D residual component is directly estimated via a neural network. Furthermore, our blur synthesizer allows for controllable blur data augmentation by modifying blur magnitude, direction, and scenes, resulting in diverse blur images. As a result, our method significantly improves deblurring performance, making it more practical for real-world scenarios.

## 1 INTRODUCTION

Motion blur is a common challenge in photography, where it is typically caused by camera or object movement during long exposure times. Given a blur image, blind motion deblurring tackles the challenge of producing a sharp image. In recent years, a breakthrough has been made by adopting diverse neural architecture designs (Nah et al., 2017; Tao et al., 2018; Kupyn et al., 2019; Cho et al., 2021; Zamir et al., 2021; Tu et al., 2022; Li et al., 2022; Chen et al., 2022; Nah et al., 2022; Zamir et al., 2022; Wang et al., 2022; Tsai et al., 2022; Li et al., 2023b; Kong et al., 2023; Fang et al., 2023; Kim et al., 2024) and utilizing the realistic blur datasets (Rim et al., 2020; 2022; Li et al., 2023a) in blind motion deblurring tasks, enabling the practical usability in real-world scenarios.

To build a more generalizable deblurring model, a realistic large-scale blur dataset containing diverse scenes and blur patterns is required. Nevertheless, existing realistic blur datasets suffer from a limited number of scenes and blur patterns because their dual camera system is heavy and complicated to collect a large-scale blur dataset (Rim et al., 2020; 2022; Li et al., 2023a). Data augmentation is an alternative to artificially increase the amount of data via a synthesis procedure. For example, one can synthesize the blur image using blur synthesizers and train a deblurring model using both real and synthetic blur images. Despite its significant successes in high-level vision tasks (DeVries & Taylor, 2017; Zhang et al., 2018; Yun et al., 2019; Cubuk et al., 2019; 2020; Hendrycks et al., 2020; Kim et al., 2021; Yang et al., 2021), there has been little study particularly focused on blur data augmentation (Zhang et al., 2020; Carbajal et al., 2023; Wu et al., 2024; Lee et al., 2024).

In fact, the previous literature on blur data augmentation exhibits certain limitations that merit further investigation: (1) diffusion-based data augmentation (Wu et al., 2024) requires ground-truth video

---
*Corresponding author

Figure 1: Our controllable blur image synthesis. We can manipulate the motion amount (amplitude), direction (phase) of the 3D-aware vector fields, and target scene images to generate a variety of blur images. They are used for data augmentation in training the deblurring model.

frame images which are not typically provided in blur datasets. Hence, its applicability to other blur datasets for blur data augmentation remains challenging. and (2) the kernel-based method (Carbajal et al., 2023) offers estimated blur kernels. However, simply regressing the simulated kernels without physically-driven blur modeling leads to unrealistic synthetic blur images which are not acceptable in real-world scenarios (Gong et al., 2017; Tran et al., 2021). Above all, the existing methods on this line neglect to pay attention to explicit 3D geometry for blur synthesis despite the fact that the camera and object motion primarily occurs in 3D space (Whyte et al., 2010). This may lead to inaccurate or unrealistic blur patterns, constraining their practical usability for blur data augmentation.

In this paper, we propose a controllable and 3D-aware blur image synthesizer to generate various realistic blur images as shown in Fig. 1, leading to better deblurring performance. Our intuition is that *intricate* 2D non-uniform motion kernels can be parameterized by a *simple* 3D camera position trajectory using rotation and translations. Such parametric technique helps reduce ill-posedness of the problem while it is closely grounded in realistic blur modeling. In the case of camera motion, we estimate 3D camera positions within the exposure time, use them for generating the transformed scene images with grid sampling (Jaderberg et al., 2015), and aggregate them to produce a synthesized blur image as shown in Fig. 2. We are motivated by the fact that the 3D camera positions projected onto the 2D image plane inherently lie in 2D space. Hence, they can be modeled by a combination of 2D rigid transformations and projected 3D residual components as shown in Fig. 2: the former is represented by *parametric* vector fields, and the latter is represented by *non-parametric* vector fields. This decomposition allows for 3D rigid transformation without depth measurements, as the 3D residual component is directly estimated via a neural network. Furthermore, our parametric and non-parametric motion modeling takes advantage of reducing ill-posedness (by parametric) and effectively capturing 3D residual components (by non-parametric).

In the case of object motion, the 3D residual components also serve as capturing object motion behaviors such as non-uniform and non-rigid deformation due to the flexibility and versatility of the non-parametric vector field. Therefore, the 3D residual components represent not only 3D *camera* residual components but also *object* residual components. Furthermore, while the non-parametric vector field is essential for capturing 3D camera motion as shown in Fig. 3 and object motion as shown in Fig. 4, its flexible nature can lead to undesirable results and ambiguity as shown in Fig. 6 (a). To overcome this, we introduce Laplacian and invertible geometric regularizations which enables us to produce more natural motions. Specifically, the 3D transformed scene image can be traced back to the original image using our invertible geometric regularization. As a result, the non-parametric vector field remains geometrically structured. Furthermore, a key aspect of our method is to represent motion as a vector field, which provides an inherently flexible framework where the magnitude and direction of each vector can be easily adjusted. This controllability allows for the generation of millions of diverse blur patterns by randomly varying the blur intensities, directions and scenes during the deblurring training, enabling the deblurring model to handle a wide range of unseen blur scenarios. We summarize our contributions as follows:

- We propose a controllable 3D-aware blur synthesizer that combines 2D motion and 3D residual components to represent 3D motion. This enables 3D transformation without depth measurements and facilitates accurate 3D motion modeling for blur data augmentation.

- Our blur synthesizer enables controllable blur data augmentation by modifying blur magnitude, direction, and scenes. It does not require any extra data such as ground-truth kernels, depths or frame images during the training, which ensures compatibility to any blur dataset.

- We demonstrate that our blur data augmentation scheme contributes to superior deblurring performance across various network architectures and blur datasets.

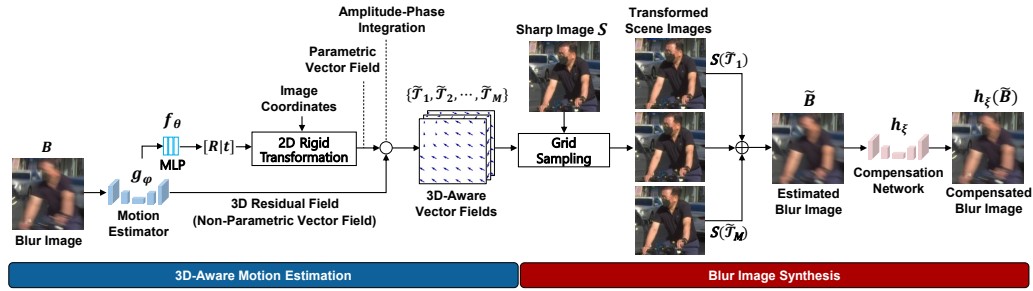

Figure 2: Training procedure for our blur synthesis model. We first predict 3D-aware vector fields (parametric + non-parametric) from a given blur image, which are used to generate corresponding scene images. These images are then aggregated to construct a realistic blur image. Note that the motion estimator, MLP, and compensation network are jointly trained.

## 2 RELATED WORKS

**Data augmentation.** Data augmentation has been widely explored in the field of image classification, where it is proven to be effective for improving the model generalization (DeVries & Taylor, 2017; Yun et al., 2019; Cubuk et al., 2019; Lim et al., 2019; Cubuk et al., 2020; Hendrycks et al., 2020; Kim et al., 2021; Yang et al., 2021). On the other hand, there has been little study particularly focused on blur data augmentation (Zhang et al., 2020; Rim et al., 2022; Wu et al., 2024). Blur Pipeline (Rim et al., 2022) presents a method to generate synthetic blur images based on GoProU dataset using saturation and noise synthesis (not data-agnostic and not controllable). Diffusion-based data augmentation (Wu et al., 2024) is a scheme to generate a synthetic blur dataset and use it for pre-training and fine-tuning the deblurring model (not cost-effective). Also, it requires the video frame images for extracting the optical flow (not data-agnostic), where the optical flow is used for the condition of the diffusion model. Also, these fail to explicitly account for the 3D geometry, despite the fact that the motion inherently occurs in 3D space (not accurate). On the other hand, our method considers the explicit 3D geometry, such that it achieves a realistic blur synthesis (*accurate*). Furthermore, our method can be *controllable*, applied to any blur dataset (*data-agnostic*), and used during the training of the deblurring model (*cost-effective*).

**Deblurring methods.** The kernel-based methods estimate blur kernels or motion information to reconstruct latent sharp images (Gong et al., 2017; Zhang et al., 2021; Carbajal et al., 2023). We may utilize these byproducts, i.e., blur kernels, to synthesize blur images. However, simply regressing the simulated kernels without physically-driven blur modeling may not hold in generating realistic blur images. In recent years, as realistic blur-sharp pair datasets (Rim et al., 2020; 2022; Li et al., 2023a) have been released, they have enabled direct prediction of latent sharp images without explicit kernel estimation. A variety of neural architecture designs (Kupyn et al., 2018; Cui et al., 2024; Mao et al., 2023; Tu et al., 2022; Cho et al., 2021; Zamir et al., 2021; Chen et al., 2022; Zamir et al., 2022; Kong et al., 2023) has emerged to improve the deblurring performance. Also, some works introduce additional self-generated priors to further improve deblurring performance (Li et al., 2022; Fang et al., 2023; Kim et al., 2024). They estimate the degradation representation (Li et al., 2022), non-uniform kernel (Fang et al., 2023), and blur segmentation map (Kim et al., 2024) as prior information and exploit them for improving the deblurring performance. These methods require additional computational costs to generate their own priors. On the other hand, our method is used for blur data augmentation during the training, such that it does not require additional inference cost.

## 3 CONTROLLABLE 3D-AWARE BLUR SYNTHESIS

### 3.1 2D BLUR SYNTHESIS MODEL

In this section, we present a new blur synthesis framework that considers a 3D perspective on blur modeling, rather than simply regressing non-uniform motion kernels from a 2D perspective. Simple regression-based motion estimation may not be generalizable and controllable. Our primary goal is to estimate camera positions in 3D space within the exposure time, generate the corresponding scene images, and aggregate them to achieve a realistic blur image synthesis. To this end, we

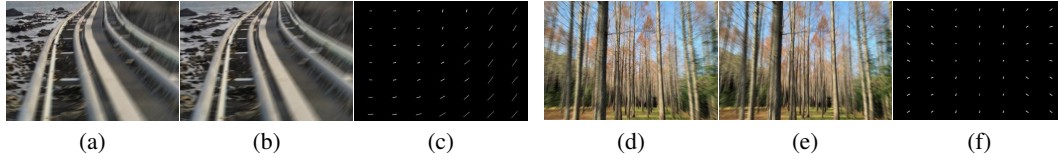

| (a) | (b) | (c) | (d) | (e) | (f) |

Figure 3: Blur synthesis results on 3D motion blur such as in-plane rotation blur, e.g., z-axis rotation: (a) Blur image, (b) Blur synthesis result, and (c) Estimated blur trajectory, and forward motion blur, e.g., z-axis translation: (d) Blur image, (e) Blur synthesis result, and (f) Estimated blur trajectory.

consider a dataset $\mathcal{D} = \{(B, S)\}$, which contains a pair of blur image $B$ and sharp image $S$. Let $\mathcal{T}_\tau = \{\mathcal{T}_\tau(\mathbf{u}_i)\}_{i=1}^N$ be a vector field, i.e., a set of transformed coordinates, where $\mathbf{u}_i \in \mathbb{R}^2$ denotes a 2D image coordinate at a pixel index $i$, $\mathcal{T}_\tau(\mathbf{u}_i) \in \mathbb{R}^2$ indicates $i$ th transformed pixel coordinate at a time $\tau$, and $N$ is the number of pixels. Note that we initially consider 2D motion and extend it to 3D motion subsequently. The blur image is expressed by using the camera exposure model over the exposure time $T$:

$$B = \int_0^T S(\mathcal{T}_\tau) \, d_\tau, \tag{1}$$

where $S(\mathcal{T}_\tau)$ is a scene image at a time $\tau$. As scenes may change due to object movements and camera shake over the exposure time, we generate the corresponding scene images $\{S(\mathcal{T}_1), \cdots, S(\mathcal{T}_M)\}$ using transformed vector field $\{\mathcal{T}_1, \cdots, \mathcal{T}_M\}$ with grid sampling (Jaderberg et al., 2015). Note that the exposure time $T$ is approximately divided into $M$ discrete camera positions for implementations. Here, we begin by considering a certain case, i.e., camera motion, which is easily encoded by a rigid transformation. The 2D rigid transformation is simply parameterized by $[\mathbf{R}_\tau | \mathbf{t}_\tau]$, e.g., rotation and translation. The transformed coordinate $\mathcal{T}_\tau(\mathbf{u})$ is denoted by

$$\mathcal{T}_\tau(\mathbf{u}) = [\mathbf{R}_\tau | \mathbf{t}_\tau]\mathbf{u} = \begin{bmatrix} r_\tau^{(11)} & r_\tau^{(12)} & t_\tau^{(1)} \\ r_\tau^{(21)} & r_\tau^{(22)} & t_\tau^{(2)} \end{bmatrix} \begin{bmatrix} x \\ y \\ 1 \end{bmatrix} = \begin{bmatrix} r_\tau^{(11)}x + r_\tau^{(12)}y + t_\tau^{(1)} \\ r_\tau^{(21)}x + r_\tau^{(22)}y + t_\tau^{(2)} \end{bmatrix}. \tag{2}$$

Since the unknown parameters are now $[\mathbf{R}|\mathbf{t}]$ in (1), one can view the blur synthesis problem as a camera pose estimation by a neural network, $f_\theta : B \to [\boldsymbol{\gamma}_1, \cdots, \boldsymbol{\gamma}_M, \mathbf{t}_1, \cdots, \mathbf{t}_M]$ where $\boldsymbol{\gamma}_\tau \in \mathbb{R}^3$ is the axis-angle representation (Murray et al., 2017) for a rotation matrix and $\mathbf{t}_\tau \in \mathbb{R}^2$ indicates a translation component. Once we estimate $\boldsymbol{\gamma}_\tau$, we can generate a 3D rotation matrix using Rodrigues' formula (Murray et al., 2017), whose 2D components are used to construct a 2D transformation matrix $\mathbf{R}_\tau$. Given $\mathbf{R}_\tau$ and $\mathbf{t}_\tau$, we can make a 2D rigid transformation field $\mathcal{T}_\tau$ using (2) and optimize the following loss to train the blur synthesis model, i.e., $L_{\texttt{blur}} = \|B - \frac{1}{M}\sum_\tau S(\mathcal{T}_\tau)\|_1$.

### 3.2 3D-AWARE BLUR SYNTHESIS MODEL

**3D camera motion blur synthesis.** We further develop our blur synthesis model, elaborating on its extension to 3D space. In fact, the camera motion appears in 3D space, but it is primarily modeled from a 2D perspective, i.e., 2D non-uniform kernels (Carbajal et al., 2023; Fang et al., 2023), which is insufficient to ensure the geometric coherence of the camera motions. Therefore, we begin by investigating the 3D rigid transformation below

$$\mathcal{T}_\tau(\mathbf{X}) = [\mathbf{R}_\tau | \mathbf{t}_\tau]\mathbf{X} = \begin{bmatrix} r_\tau^{(11)} & r_\tau^{(12)} & r_\tau^{(13)} & t_\tau^{(1)} \\ r_\tau^{(21)} & r_\tau^{(22)} & r_\tau^{(23)} & t_\tau^{(2)} \\ r_\tau^{(31)} & r_\tau^{(32)} & r_\tau^{(33)} & t_\tau^{(3)} \end{bmatrix} \begin{bmatrix} x \\ y \\ z \\ 1 \end{bmatrix} = \begin{bmatrix} r_\tau^{(11)}x + r_\tau^{(12)}y + r_\tau^{(13)}z + t_\tau^{(1)} \\ r_\tau^{(21)}x + r_\tau^{(22)}y + r_\tau^{(23)}z + t_\tau^{(2)} \\ r_\tau^{(31)}x + r_\tau^{(32)}y + r_\tau^{(33)}z + t_\tau^{(3)} \end{bmatrix}, \tag{3}$$

where $\mathbf{X}$ is a 3D canonical coordinate. We discover that the 3D rigid transformation vector $\mathcal{T}_\tau(\mathbf{X}) \in \mathbb{R}^3$ can be decomposed into 2D rigid transformation vector $\mathcal{T}_\tau^*(\mathbf{u}) = [\mathcal{T}_\tau(\mathbf{u}), 0] \in \mathbb{R}^3$ and 3D residual vector $\mathcal{E}_\tau(\mathbf{X}) \in \mathbb{R}^3$, written by

$$\mathcal{T}_\tau(\mathbf{X}) = \underbrace{\begin{bmatrix} r_\tau^{(11)}x + r_\tau^{(12)}y + t_\tau^{(1)} \\ r_\tau^{(21)}x + r_\tau^{(22)}y + t_\tau^{(2)} \\ 0 \end{bmatrix}}_{\mathcal{T}_\tau^*(\mathbf{u})} + \underbrace{\begin{bmatrix} r_\tau^{(13)}z \\ r_\tau^{(23)}z \\ r_\tau^{(31)}x + r_\tau^{(32)}y + r_\tau^{(33)}z + t_\tau^{(3)} \end{bmatrix}}_{\mathcal{E}_\tau(\mathbf{X})}. \tag{4}$$

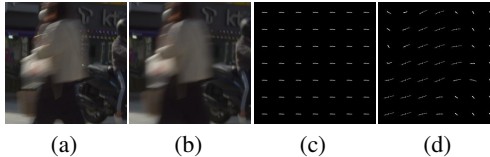 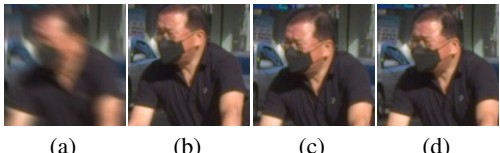

(a) (b) (c) (d)  (a) (b) (c) (d)

Figure 4: Visual results on (a) Blur image, (b) Synthesized blur image, (c) 2D parametric blur trajectory, and (d) 2D parametric + 3D non-parametric blur trajectory. Person movements are captured by (d), but not (c).

Figure 5: Visual results on (a) Synthesized blur image, (b-d) transformed scene images. Our blur synthesizer generates 3D-aware transformed scene images without requiring ground-truth video frame images.

Using a projection operation $\pi : \mathbb{R}^3 \rightarrow \mathbb{R}^2$ with camera intrinsics $K$, we compute the projected coordinate vector $\tilde{\mathcal{T}}_\tau(\mathbf{u}) = \pi(\mathcal{T}_\tau(\mathbf{X}); K) = \pi(\mathcal{T}_\tau^*(\mathbf{u}) + \mathcal{E}_\tau(\mathbf{X}); K)$ to sample 3D-aware transformed scene images over the exposure time. Here, we are motivated by the fact that $\tilde{\mathcal{T}}_\tau(\mathbf{u})$ lies in $\mathbb{R}^2$. Therefore, following the rationale of the decomposition principle in (4), we express the projected coordinate vector $\tilde{\mathcal{T}}_\tau(\mathbf{u})$ as a combination of 2D rigid transformation coordinate $\mathcal{T}_\tau(\mathbf{u}) \in \mathbb{R}^2$ and projected 3D residual vector[1] $\boldsymbol{\epsilon}_\tau(\mathbf{u}) \in \mathbb{R}^2$ derived from $\mathcal{E}_\tau(\mathbf{X})$, i.e.,

$$\tilde{\mathcal{T}}_\tau(\mathbf{u}) = \mathcal{C}(\mathcal{T}_\tau(\mathbf{u}), \boldsymbol{\epsilon}_\tau(\mathbf{u})), \tag{5}$$

where $\mathcal{C}$ serves as a composition function for integrating the two components. Since $\tilde{\mathcal{T}}_\tau(\mathbf{u})$ contains 3D residual components, it is referred to as the 3D-aware coordinate vector. Then, we simply induce the 3D-aware vector field, i.e., $\tilde{\mathcal{T}}_\tau = \mathcal{C}(\mathcal{T}_\tau, \boldsymbol{\epsilon}_\tau)$ where $\mathcal{T}_\tau$ is the 2D rigid transformation field as discussed in Section 3.1 and $\boldsymbol{\epsilon}_\tau$ denotes the 3D residual field modeled by a neural network, $g_\varphi : B \rightarrow \{\boldsymbol{\epsilon}_1, \boldsymbol{\epsilon}_2, \cdots, \boldsymbol{\epsilon}_M\}$. This decomposition scheme allows for 3D rigid transformation without requiring explicit depth measurements[2], as the 3D residual field is directly estimated via the neural network. To investigate the composition function $\mathcal{C}$ for better motion controllability, we first reformulate the 3D-aware vector field using a known canonical vector field $\mathbf{U}$, i.e., $S = S(\mathbf{U})$, which is

$$\tilde{\mathcal{T}}_\tau = \mathcal{C}(\mathcal{T}_\tau, \boldsymbol{\epsilon}_\tau) = \mathcal{C}(\underbrace{\mathbf{U} + \Delta\mathcal{T}_\tau}_{\mathcal{T}_\tau}, \boldsymbol{\epsilon}_\tau) = \mathbf{U} + \underbrace{\mathcal{C}(\Delta\mathcal{T}_\tau, \boldsymbol{\epsilon}_\tau)}_{\boldsymbol{\delta}_\tau} = \mathbf{U} + \boldsymbol{\delta}_\tau, \tag{6}$$

where $\boldsymbol{\delta}_\tau = \mathcal{C}(\Delta\mathcal{T}_\tau, \boldsymbol{\epsilon}_\tau)$ is a displacement field which consists of the 2D rigid transformation residual field $\Delta\mathcal{T}_\tau$ (*parametric* field) and 3D residual field $\boldsymbol{\epsilon}_\tau$ (*non-parametric* field). Note that the canonical vector field $\mathbf{U}$ is the constant field, such that it is factored out from the composition function. Then, we adopt amplitude-phase integration in polar coordinates for the composition function, i.e., $\boldsymbol{\delta}_\tau = \mathcal{C}(\Delta\mathcal{T}_\tau, \boldsymbol{\epsilon}_\tau) = |\Delta\mathcal{T}_\tau| \cdot |\boldsymbol{\epsilon}_\tau| \angle(\phi(\Delta\mathcal{T}_\tau) + \phi(\boldsymbol{\epsilon}_\tau))$, where $|\cdot|$ indicates the magnitude of the vector and $\phi$ is a function to calculate the vector angle. This integration scheme provides a straight-forward way to manipulate the magnitude and direction of the motion, facilitating a controllable blur synthesis, as opposed to simple vector field integration, i.e., $\boldsymbol{\delta}_\tau = \mathcal{C}(\Delta\mathcal{T}_\tau, \boldsymbol{\epsilon}_\tau) = \Delta\mathcal{T}_\tau + \boldsymbol{\epsilon}_\tau$. Given a target scene image $S$ and the 3D-aware vector field $\tilde{\mathcal{T}}_\tau$, we can render 3D-aware transformed scene images and synthesize a blur image as exemplified in Fig. 5 by

$$\tilde{B} = \frac{1}{M} \sum_\tau S(\tilde{\mathcal{T}}_\tau). \tag{7}$$

Finally, we suggest minimizing the following loss to train the 3D-aware blur synthesis model:

$$L_{\texttt{blur\_3D}} = \|B - \tilde{B}\|_1 + \|B - h_\xi(\tilde{B})\|_1, \tag{8}$$

where $h_\xi$ is an auxiliary network that compensates for photometric variations between blur and sharp images, arising from different image sensors, lenses, and color drifts. This compensation is required to encourage the 3D-aware vector fields to focus only on blur components (not disrupted by the photometric issues). This will be more discussed in Section C of Appendix. Furthermore, we emphasize that our compositional training framework leverages structured parametric modeling

---

[1] We provide underlying derivation and insights of projected 3D residual vector in Section A of Appendix.

[2] In our experiments, we observe that inaccurate absolute depth measurements can lead to performance degradation, as discussed in detail in Section 4.4.

to alleviate ill-posedness and flexible non-parametric modeling to effectively capture 3D residual components, thereby achieving accurate 3D motion modeling.

**3D object motion blur synthesis.** As discussed earlier, we estimate the 3D residual components using the non-parametric vector field $\epsilon_\tau$, which is initially designed to represent the 3D *camera* residual fields. Additionally, the 3D residual components are capable of representing *object* residual fields. In fact, non-parametric motion modeling, i.e., directly estimating the vector field via a neural network, is particularly effective in capturing the object motion behaviors (i.e., non-uniform and non-rigid motion) due to its flexibility and versatility. Hence, our 3D-aware blur synthesis model (7) is inherently compatible with scenarios containing object motion without further modification. As a result, our method can handle 3D camera motions as shown in Fig. 3 and object motions as shown in Fig. 4, using (6). Additional visualization results can be found in Fig. 14 and 15 of Appendix.

### 3.3 Ambiguity regularization and total loss

As discussed in Section 3.2, the non-parametric vector field provides substantial flexibility, but its arbitrary and unconstrained natures can lead to ambiguities. This makes the optimization more challenging. As a result, it yields implausible synthesis images and artifacts as shown in Fig. 6 (a). To address this, we present Laplacian regularization and invertible geometric regularization. These regularizations help mitigate the ambiguities and ensure the geometric coherence regarding the non-parametric vector field.

**Laplacian regularization.** The Laplacian regularization helps smooth irregular vector fields. This holds under the prior knowledge that the motion vector field is not irregularly changed in the spatial space. To promote the smoothness of the vector fields, we derive the Laplacian regularization with respect to the 3D-aware vector field $\tilde{\mathcal{T}}_\tau$:

$$L_{\text{smooth}} = \sum_\tau \sum_{x,y} \left( 4\tilde{\mathcal{T}}_\tau(x,y) - \sum_{i,j} \tilde{\mathcal{T}}_\tau(x+i,y) + \tilde{\mathcal{T}}_\tau(x,y+j) \right)^2, \qquad i,j \in \{-1,1\}. \quad (9)$$

**Invertible geometric regularization.** To adapt the geometric consistency for the 3D-aware vector field $\tilde{\mathcal{T}}_\tau$, we propose a new invertible geometric regularization that allows us to trace back to the original scene image. Specifically, the 3D transformed scene image $\tilde{S}_\tau = S(\tilde{\mathcal{T}}_\tau)$ should be geometrically reverted to the original one $S$ via an inverse vector field. To this end, we recall the 3D-aware vector field (6), i.e., $\tilde{\mathcal{T}}_\tau = \mathbf{U} + \boldsymbol{\delta}_\tau$, where $\boldsymbol{\delta}_\tau$ denotes the displacement field. The inverse vector field $\tilde{\mathcal{T}}'_\tau$ is computed by using the reversed direction of the displacement field, i.e., $\tilde{\mathcal{T}}'_\tau = \mathbf{U} - \boldsymbol{\delta}_\tau$. Finally, the geometric-consistent vector field is achieved by optimizing the following loss:

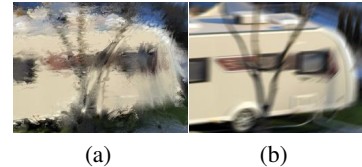

(a)          (b)

Figure 6: Synthesized blur images (a) without regularizations and (b) with regularizations.

$$L_{\text{geometric}} = \frac{1}{M} \sum_\tau \|S - \tilde{S}_\tau(\tilde{\mathcal{T}}'_\tau)\|_1, \quad (10)$$

where $\tilde{S}_\tau(\tilde{\mathcal{T}}'_\tau)$ is the geometrically inverted scene image. This geometric consistency loss is the key component of our method to ensure that the 3D-aware vector field remains geometrically structured. Therefore, it ensures geometric-aware blur synthesis as shown in Fig. 6 (b).

**Total loss.** We suggest minimizing the following loss to train our geometric-aware blur synthesizer:

$$L_{\text{total}} = L_{\text{blur\_3D}} + \lambda_1 L_{\text{smooth}} + \lambda_2 L_{\text{geometric}}, \quad (11)$$

where $\lambda_1, \lambda_2 > 0$ are hyper-parameters to balance the loss terms, which will be discussed in Section 4.5 and Section F.1 of Appendix.

### 3.4 Controllable blur image synthesis

Our blur synthesis model is designed to enable blur data augmentation during the deblurring model training, thereby improving final deblurring performance and model generalization. Specifically, if

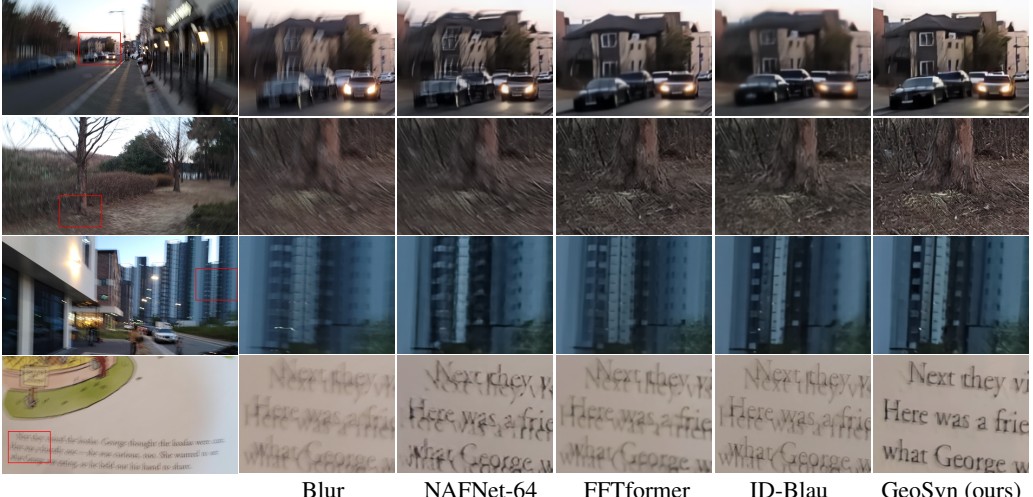

Blur     NAFNet-64     FFTformer     ID-Blau     GeoSyn (ours)

Figure 7: Qualitative comparison results on real-world blur images. NAFNet is used for all methods. Our blur synthesizer produces sharper and artifact-free results compared to state-of-the-art methods.

a blur dataset contains $3,000$ blur images and the deblurring model is trained over $1,000$ epochs, it generates approximately 3,000,000 different synthetic blur images. This extensive diversity enhances the generalization ability of the deblurring model, enabling it to handle a wide range of unseen blur scenarios. To this end, we first extract a 3D-aware displacement vector $\boldsymbol{\delta} = (x_\delta, y_\delta)$ from (6). Note that we use the notation $\boldsymbol{\delta}$ for simplicity here. It can be converted into the amplitude-phase representations in the polar coordinates, i.e., $\boldsymbol{\delta} = |\boldsymbol{\delta}| \angle \phi(\boldsymbol{\delta})$ where $|\boldsymbol{\delta}| = \sqrt{x_\delta^2 + y_\delta^2}$ indicates the amplitude of the vector, i.e., motion intensity and $\phi(\boldsymbol{\delta}) = \tan^{-1}(\frac{y_\delta}{x_\delta})$ represents the phase of the vector, i.e., motion direction. Then, every displacement vector is controlled by the amplitude control parameter $\alpha$ and phase control parameter $\beta$, i.e., $\tilde{x_\delta} = \alpha|\boldsymbol{\delta}|\cos(\phi(\boldsymbol{\delta}) + \beta)$ and $\tilde{y_\delta} = \alpha|\boldsymbol{\delta}|\sin(\phi(\boldsymbol{\delta}) + \beta)$, whose parameters are explored in Section B of Appendix. We remark that this amplitude-phase motion adjustment preserves the geometric structure of the vector fields since the control parameters are identically applied to all vector fields generated from a single blur image to adjust overall blur intensity and direction. Meanwhile, the control parameters can be determined for each blur image. Finally, we employ this modified displacement motion vector, $\tilde{\boldsymbol{\delta}} = (\tilde{x_\delta}, \tilde{y_\delta})$ instead of the original one $\boldsymbol{\delta}$ for the subsequent scene image rendering and blur image synthesis as in (7). We present some blur synthesis results for different scenes, amplitudes, and phases as shown in Fig. 1, Fig. 16 and 17 of Appendix.

## 4 EXPERIMENTS

### 4.1 EXPERIMENTAL SETUP

**Datasets and evaluation metrics.** For datasets, we use RealBlur (Rim et al., 2020), RSBlur (Rim et al., 2022), and GoPro (Nah et al., 2017) datasets for training and evaluation. RealBlur is split into two types: RealBlur-J (sRGB domain) and RealBlur-R (RAW domain). Each RealBlur type consists of 3,758 and 980 image pairs for training and test sets, respectively. RSBlur contains 8,878 and 3,360 blur-sharp image pairs for training and test sets, respectively. GoPro contains 2,103 and 1,111 image pairs for training and test sets, respectively. We use the evaluation metrics such as Peak Signal to Noise Ratio (PSNR) and Structural SIMilarity (SSIM) (Wang et al., 2004).

**Implementation details.** We present the implementation details for training our blur synthesis model. Our blur synthesis model consists of a motion estimator (NAFNet-8) and compensation network (NAFNet-16). NAFNet-16 means NAFNet (Chen et al., 2022) with 16 base channel widths. Note that the compensation network is only used for the photometric compensation during the training, as discussed in Section C of Appendix. According to Fig. 2, the blur input image is fed into the encoder of the motion estimator. Then, the intermediate features are fed to MLP, including global average pooling, two fully connected layers, and one ReLU activation, in order to estimate 2D motion parameters $\{\boldsymbol{\gamma}_1, \cdots, \boldsymbol{\gamma}_M, \mathbf{t}_1, \cdots, \mathbf{t}_M\}$. Using the parameters, we obtain parametric vector

Table 1: Comparison results across diverse neural networks and blur datasets.

| Model | GoPro | | HIDE | | RealBlur-J | | RealBlur-R | |
|---|---|---|---|---|---|---|---|---|
| | PSNR | SSIM | PSNR | SSIM | PSNR | SSIM | PSNR | SSIM |
| MIMO-UNet+ | 32.44 | 0.957 | 30.00 | 0.930 | 31.92 | 0.919 | 39.10 | 0.969 |
| + GeoSyn (ours) | **33.01** | **0.962** | **30.86** | **0.940** | **32.55** | **0.925** | **39.68** | **0.972** |
| Restormer | 32.92 | 0.961 | 31.22 | 0.942 | 32.32 | 0.924 | 39.47 | 0.972 |
| + GeoSyn (ours) | **33.37** | **0.964** | **31.61** | **0.946** | **33.05** | **0.937** | **40.31** | **0.974** |
| NAFNet | 33.69 | 0.966 | 31.32 | 0.943 | 32.50 | 0.928 | 39.89 | 0.973 |
| + GeoSyn (ours) | **34.09** | **0.969** | **31.64** | **0.947** | **32.99** | **0.936** | **40.49** | **0.976** |
| FFTformer | 34.21 | 0.969 | 31.62 | 0.946 | 32.62 | 0.933 | 40.11 | 0.973 |
| + GeoSyn (ours) | **34.39** | **0.970** | **31.98** | **0.949** | **33.68** | **0.938** | **40.89** | **0.977** |

fields $\{\mathcal{T}_1, \mathcal{T}_2, \cdots, \mathcal{T}_M\}$ as discussed in Section 3.1. Also, the intermediate features are fed into the decoder of the motion estimator to predict non-parametric vector fields $\{\boldsymbol{\epsilon}_1, \boldsymbol{\epsilon}_2, \cdots, \boldsymbol{\epsilon}_M\}$. Then, we aggregate the parametric and non-parametric vector fields using the amplitude-phase integration and finally obtain the 3D-aware vector fields as discussed in Section 3.2. To produce diverse blur images for blur data augmentation, we modify amplitudes, phases of the vector field, and scene contents as discussed in Section 3.4. We will discuss how to synthesize and where to synthesize in detail in Section B of Appendix. Note that our blur synthesizer is only used for training. Therefore, it does not increase the computation cost in the evaluation stage. We use blur datasets randomly cropped by $256 \times 256$ during the training. We train our blur synthesizer up to $1,000$ epochs for RealBlur, $3,000$ epochs for GoPro, and $500$ epochs for RSBlur. Also, our blur synthesizer is optimized by the AdamW (Loshchilov & Hutter, 2019) algorithm ($\beta_1 = 0.9$, $\beta_2 = 0.9$ and weight decay $1e^{-3}$) with the cosine annealing schedule ($1e^{-3}$ to $1e^{-7}$) (Loshchilov & Hutter, 2016) gradually reduced for total iterations of each dataset. Unless otherwise specified, we use $\lambda_1 = 0.1$, $\lambda_2 = 1.0$ and the number of camera positions as 16, which is discussed in Section 4.5 and F.1 of Appendix. We use RealBlur-J for all ablation studies. The pseudo-codes can be found in Section G and H of Appendix.

## 4.2 Main results

To demonstrate the effectiveness of our blur synthesizer, we conduct experiments across various datasets such as GoPro (Nah et al., 2017), HIDE (Shen et al., 2019), RealBlur-J and R (Rim et al., 2020), and network architectures such as MIMO-UNet+ (Cho et al., 2021), Restormer (Zamir et al., 2022), NAFNet (Chen et al., 2022) and FFTformer (Kong et al., 2023). We train with GoPro and evaluate GoPro and HIDE. Also, we train with RealBlur-J and R, and evaluate the corresponding trained models. As shown in Table 1, the results clearly demonstrate that our blur synthesizer, i.e., GeoSyn boosts deblurring performance across not only different network architectures but also various datasets. For example, FFTformer equipped with our GeoSyn significantly improves PSNR from 32.62 to 33.68 dB on RealBlur-J. Furthermore, we present a more comprehensive analysis of the generalization ability of our GeoSyn in Section E. Also, we present the visual comparison results on GoPro as shown in Fig. 12 of Appendix, RealBlur as shown in Fig. 13 of Appendix, and real-world blur images as shown in Fig. 7 and Fig. 11 of Appendix.

## 4.3 Comparison to other methods

**Comparison to kernel-based methods.** J-MKPD (Carbajal et al., 2023) and MotionETR (Zhang et al., 2021) generate motion information for deblurring purposes. However, motion information can be used for blur data augmentation. Since their motion estimators are trained with GoPro, we train our blur synthesizer with GoPro for a fair comparison. Then, the deblurring model, NAFNet-64 is trained with GoPro, using data augmentation with their motion estimators and ours. As shown in the upper side of Table 2, MotionETR exhibits a performance improvement from 33.69 to 33.92 dB, but it lags behind our GeoSyn (34.09 dB). We believe their motion information does not account for explicit 3D geometry, causing performance limitations.

**Comparison to blur synthesis methods.** Blur Pipeline (Rim et al., 2022) and ID-Blau (Wu et al., 2024) are the blur data augmentation methods and are trained with a certain dataset (GoPro). For example, ID-Blau requires the ground-truth video frame images to extract optical flows, such that it

Table 2: Comparison results against related works. We compare our method with kernel-based methods (above) and blur data augmentation methods (below).

| Augmentation Types | Datasets | PSNR ↑ | SSIM ↑ |
|---|---|---|---|
| None | | 33.69 | 0.966 |
| J-MKPD | GoPro | 33.59 | 0.965 |
| MotionETR | | 33.92 | 0.968 |
| GeoSyn (ours) | | **34.09** | **0.969** |
| None | | 32.50 | 0.928 |
| Blur Pipeline | | 32.57 | 0.931 |
| ID-Blau | RealBlur-J | 32.70 | 0.932 |
| GeoSyn (ours) | | **32.99** | **0.936** |
| ID-Blau + GeoSyn (ours) | | **33.09** | **0.938** |

Table 3: Comparison results on RSBlur. The best results are indicated in bold.

| Methods | GMACs | RSBlur | |
|---|---|---|---|
| | | PSNR ↑ | SSIM ↑ |
| SRN-Deblur | 1434.82 | 32.53 | 0.840 |
| MIMO-UNet+ | 154.41 | 33.37 | 0.856 |
| MPRNet | 777.01 | 33.61 | 0.861 |
| Restormer | 141.00 | 33.69 | 0.863 |
| Uformer-B | 89.50 | 33.98 | 0.866 |
| ConvIR-L | 71.22 | 34.06 | 0.868 |
| NAFNet-64 | 63.64 | 33.97 | 0.866 |
| + GeoSyn (ours) | 63.64 | **34.23** | **0.870** |
| SegDeblur-L | 62.68 | 34.21 | 0.870 |
| + GeoSyn (ours) | 62.68 | **34.31** | **0.872** |

Table 4: Ablation study on various types of vector fields. "P" indicates the parametric vector field while "NP" exhibits the non-parametric vector field. The best results are indicated in bold.

| Methods | P | NP | PSNR ↑ | SSIM ↑ |
|---|---|---|---|---|
| No Augmentation | | | 32.50 | 0.928 |
| 2D Parametric | ✓ | | 32.65 | 0.932 |
| 3D Non-Parametric | | ✓ | 32.61 | 0.930 |
| 3D Flat Depth | ✓ | | 32.66 | 0.932 |
| 3D Monocular Depth | ✓ | | 32.47 | 0.929 |
| GeoSyn (ours) | ✓ | ✓ | **32.99** | **0.936** |

Table 5: Ablation study on model sizes. Our blur augmentation enables us to build an efficient deblurring model. It reduces the computational cost by up to 4×.

| Methods | GMACs | PSNR ↑ | SSIM ↑ |
|---|---|---|---|
| NAFNet-16 | 4.0 | 31.58 | 0.912 |
| + GeoSyn (ours) | | **31.97** | **0.921** |
| NAFNet-32 | 16.0 | 31.99 | 0.920 |
| + GeoSyn (ours) | | **32.59** | **0.931** |
| NAFNet-64 | 63.5 | 32.50 | 0.928 |
| + GeoSyn (ours) | | **32.99** | **0.936** |

can not be trained with other datasets containing only blur and sharp images such as RealBlur (Rim et al., 2020) and RSBlur (Rim et al., 2022). As shown in the lower side of Table 2, ID-Blau, trained with GoPro, leads to limited deblurring performance improvement on RealBlur-J, from 32.50 to 32.70 dB. On the other hand, our GeoSyn can be trained with RealBlur-J, to capture dataset-specific motion patterns. Hence, our GeoSyn yields better deblurring performance (32.99 dB), thanks to our data-agnostic motion modeling. Furthermore, we remark that our GeoSyn is compatible with ID-Blau. Specifically, we train the deblurring model initialized by ID-Blau pre-trained model, with our GeoSyn. We observe that it further improves the deblurring performance (33.09 dB), achieving the best performance. More comparison results with ID-Blau can be found in Section D of Appendix.

**Results on RSBlur.** We experiment with an additional dataset, RSBlur (Rim et al., 2022) which contains a variety of camera and object motions with high-resolution images. We train and evaluate NAFNet-64 (Chen et al., 2022) and SegDeblur-L (Kim et al., 2024) with RSBlur, using our data augmentation scheme. As shown in Table. 3, our GeoSyn improves the performance on NAFNet-64, from 33.97 to 34.23 dB. Interestingly, our GeoSyn can be compatible with the prior-based method such as SegDeblur-L, improving the deblurring performance from 34.21 to 34.31 dB.

## 4.4 VARIOUS FORMS OF VECTOR FIELDS

The goal of this experiment is to identify an optimal configuration of the vector fields for blur data augmentation, by evaluating various combinations of parametric and non-parametric vector fields including 2D motion, 3D motion, and depth information, as reported in Table 4.

**2D parametric vector field.** The parametric vector field approach is beneficial for reducing the ill-posedness of the problem, rather than naïvely estimating non-parametric vector fields. However, the 2D parametric vector field that relies solely on 2D transformations fails to incorporate 3D motion information, leading to a sub-optimal performance of 32.65 dB.

**3D non-parametric vector field.** The non-parametric vector field is specialized to capture 3D motions. However, when the non-parametric method is used only, it struggles to effectively learn motion behaviors due to huge ill-posedness, e.g., the presence of numerous feasible solutions, achieving insufficient performance (32.61 dB).

**3D parametric vector field using depth information.** The parametric vector fields using 3D transformation with depth measurements yield sub-optimal results. The deblurring result (32.47 dB)

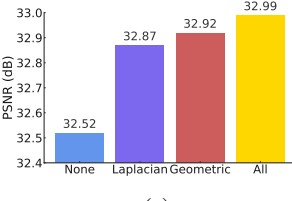 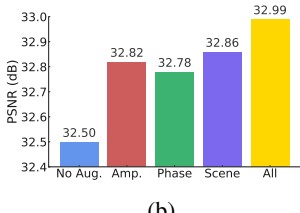 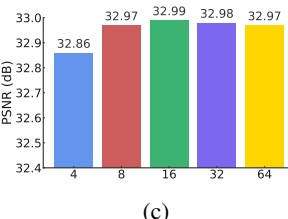

Figure 8: Ablation studies on (a) Regularizations, (b) Controllability, and (c) # of camera positions.

using the monocular depth estimates generated by Ke et al. (2024) is even worse than that of flat depth (32.66 dB). We believe that monocular depth values are inherently inaccurate because they are typically relative depth information rather than accurate absolute depth measurements required for 3D transformations, leading to unfavorable blur synthesis.

**2D parametric + 3D non-parametric vector field (ours).** Our method fuses 2D parametric and 3D non-parametric vector fields, which circumvents the need for absolute depth values. Therefore, we take advantage of both reducing ill-posedness (by parametric) and effectively modeling 3D residual components (by non-parametric), achieving the best performance (32.99 dB).

## 4.5 ABLATION STUDY

**Efficient deblurring models.** In this section, we discuss the effectiveness of our blur data augmentation scheme in constructing an efficient deblurring model. We train and evaluate NAFNet with different model sizes, incorporating our data augmentation scheme. As shown in Table. 5, the results show that our GeoSyn reduces the computational cost, i.e., GMACs, by up to $4\times$. Specifically, NAFNet-32 equipped with our data augmentation (16 GMACs, PSNR 32.59 dB) produces even better performance than that of NAFNet-64 (63.5 GMACs, PSNR 32.50 dB). This highlights the potential benefits of our blur synthesis model in building an efficient deblurring model.

**Effects on regularization.** To verify that the ambiguity regularization losses are necessary, we train deblurring models using our blur synthesizers trained without regularization, with Laplacian only ($\lambda_1 = 0.1$), with geometric only ($\lambda_2 = 1.0$), and with both regularizations. As shown in Fig. 8 (a), we observe that each regularization technique contributes to reducing ambiguities, leading to better subsequent deblurring performance ($32.50 \rightarrow 32.87$ or $32.92$ dB). Furthermore, the results demonstrate that both regularizations are crucial for achieving further improvements, reaching 32.99 dB. We explore the hyperparameters $\{\lambda_1, \lambda_2\}$ for more details in Section F.1 of Appendix.

**Controllability.** We explore the controllability of our GeoSyn. As discussed in Section 3.4, we can manipulate amplitude, phase of the vector fields, and scene contents to produce blur images with diverse blur patterns and scene contents. To confirm which augmentation type is more effective, we train deblurring models with amplitude, phase, scene, and all augmentations. As shown in Fig. 8 (b), each augmentation type demonstrates its effectiveness in improving deblurring performance. Moreover, the results indicate that all augmentation types are necessary to achieve the best performance.

**The number of camera positions.** We investigate the effect of the number of camera positions when synthesizing a blur image. We train our blur synthesizer with different numbers of camera positions: $\{4, 8, 16, 32, 64\}$, and the deblurring model using individual synthesizers. As shown in Fig. 8 (c), a sufficient number of camera positions, e.g., more than $8$, can capture intrinsic motion behaviors, such that it results in meaningful subsequent deblurring performance.

**More ablation studies.** For further insights, we perform an in-depth analysis of our blur synthesizer in Section F of Appendix.

## 5 CONCLUSIONS

In this paper, we propose a new 3D-aware blur synthesizer designed to generate diverse blur images for data augmentation, improving deblurring performance. We integrate parametric and non-parametric vector fields to take advantage of reducing the ill-posedness and modeling 3D camera and object residual components. We demonstrate the effectiveness of our blur synthesizer on various network architectures and datasets. Our 3D-aware vector fields can be applied to optical flow and depth prediction from blur images while facilitating motion estimation for video generation.

## ACKNOWLEDGEMENT

This work was supported by Institute for Information & communications Technology Promotion(IITP) grant funded by the Korea government(MSIT) (No.RS-2019-II190075, Artificial Intelligence Graduate School Program(KAIST); No.RS-2021-II212068, Artificial Intelligence Innovation Hub).

## REPRODUCIBILITY STATEMENT

To facilitate reproducibility, we present comprehensive implementation details for training our blur synthesizer in Section 4.1 and Fig. 2. We also describe the details of the data augmentation strategy, i.e., where to synthesize, how to synthesize, and how to augment during the training of the deblurring model, in Section B of Appendix. The hyperparameter settings are empirically optimized as described in Section 4.5 and Section F.1 of Appendix, and are summarized in Section 4.1. We also provide the pytorch pseudo-codes of our blur synthesis and blur data augmentation in Section G and H of Appendix.

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

## A    MORE DETAILS ON PROJECTED 3D RESIDUAL VECTOR

To derive the projected 3D residual vector, we recall the 3D transformation vector (4) as follows:

$$\mathcal{T}_\tau(\mathbf{X}) = \begin{bmatrix} X \\ Y \\ Z \end{bmatrix} = \underbrace{\begin{bmatrix} r_\tau^{(11)}x + r_\tau^{(12)}y + t_\tau^{(1)} \\ r_\tau^{(21)}x + r_\tau^{(22)}y + t_\tau^{(2)} \\ 0 \end{bmatrix}}_{R} + \underbrace{\begin{bmatrix} r_\tau^{(13)}z \\ r_\tau^{(23)}z \\ r_\tau^{(31)}x + r_\tau^{(32)}y + r_\tau^{(33)}z + t_\tau^{(3)} \end{bmatrix}}_{\mathcal{E}}. \quad (12)$$

where $R = [R_x, R_y, 0]$ is the 2D transformation vector and $\mathcal{E} = [\mathcal{E}_x, \mathcal{E}_y, \mathcal{E}_z]$ is the 3D residual vector. Note that we use the notations of $R$ and $\mathcal{E}$ instead of $\mathcal{T}_\tau^*(\mathbf{u})$ and $\mathcal{E}_\tau(\mathbf{X})$ in (4) for simplicity here. Using a projection operation $\pi : \mathbb{R}^3 \rightarrow \mathbb{R}^2$ with camera intrinsics $K$, we can compute the projected coordinate vector $\tilde{\mathcal{T}}_\tau(\mathbf{u}) = [x, y] = \pi(\mathcal{T}_\tau(\mathbf{X}); K) = \pi(R + \mathcal{E}; K)$. By using the pinhole camera principle, we express a $x$ component of the projected coordinate vector as

$$x = f_x \frac{X}{Z} + P_x = f_x \frac{R_x + \mathcal{E}_x}{\mathcal{E}_z} + P_x, \quad (13)$$

where $R_x$ is a 2D transformation component, $\mathcal{E}_x$ and $\mathcal{E}_z$ are 3D residual components, $f_x$ is a focal length, and $P_x$ is a principal point. Here, we emphasize that $x$ is characterized by the 2D transformation component $R_x$ and 3D residual components $\{\mathcal{E}_x, \mathcal{E}_z\}$. Since we define $x$ as a combination of 2D transformation component $R_x$ and projected 3D residual component $\epsilon_x$, i.e., $x = \mathcal{C}(R_x, \epsilon_x)$ where $\mathcal{C}$ is a composition function for integrating the two components, the projected 3D residual component $\epsilon_x$ contains the 3D residual components $\{\mathcal{E}_x, \mathcal{E}_z\}$, focal length $f_x$, and principal point $P_x$. These components are combined by a single value $\epsilon_x$ which is estimated by our motion estimator.

## B    DATA AUGMENTATION STRATEGY

**Where to synthesize?** The camera motion is regarded as a global motion while the object motion is treated as a local motion. To accommodate the synthesized local motion, we use CutSyn strategy, whose methodology is similar to other data augmentation schemes such as CutBlur (Yoo et al., 2020), CutMix (Yun et al., 2019) and CutOut (DeVries & Taylor, 2017). Specifically, we randomly choose a region for augmentation and apply a controllable blur synthesis, i.e., $B_{\text{aug}} = M \odot \tilde{B} + (1 - M) \odot S$ where $M \in \{0, 1\}$ is the mask image indicating where to synthesize, $\tilde{B}$ means the synthesized blur image, $S$ is the sharp image, and $\odot$ denotes the element-wise multiplication. Note that we will discuss our amplitude and phase augmentation policy, i.e., how to construct the synthesized blur image $\tilde{B}$, in the following paragraph. On the other hand, RealBlur has only camera motion blur images (global motion only). In this case, we apply the controllable blur synthesis in the whole region, e.g., $B_{\text{aug}} = \tilde{B}$. As shown in Table 6, the local data augmentation is beneficial for GoPro which contains camera and object motions. Meanwhile, the global data augmentation shows better performance on RealBlur-J since it contains camera motion only.

Table 6: Comparison results on regions of blur augmentation. The best results are indicated in bold.

| Region of augmentation | GoPro | | RealBlur-J | |
|---|---|---|---|---|
| | PSNR↑ | SSIM↑ | PSNR↑ | SSIM↑ |
| None | 33.69 | 0.966 | 32.50 | 0.928 |
| Global | 34.02 | **0.969** | **32.99** | **0.936** |
| Local (CutSyn) | **34.09** | **0.969** | 32.94 | **0.936** |

**How to synthesize?** We discuss our amplitude and phase adjustment policy for constructing a synthesized blur image $\tilde{B}$ during the deblurring training. As discussed in Section 3.4, we introduce the amplitude control parameter $\alpha$ and phase control parameter $\beta$ to adjust the amplitude and phase of the displacement field, i.e., $\tilde{\boldsymbol{\delta}} = \alpha|\boldsymbol{\delta}|\angle(\phi(\boldsymbol{\delta}) + \beta)$. Since many data augmentation approaches adopt a random augmentation policy and show remarkable performance, we also use the random choice of both amplitude and phase control parameters. To determine the dynamic range of amplitude and phase control parameters for random augmentations, we conduct experiments on various ranges, as presented in Table 7. For phase augmentation, although the best performance is observed across multiple phase ranges, we select the range of -90° to 90°, as it allows for phase augmentation over a

wider range of phase variations. For amplitude augmentation, the bounded amplitude augmentation gives the best performance. Specifically, the dynamic range of the phase values is bounded between -180° and 180° even though adjusting the phase values by $\beta$. Meanwhile, the dynamic range of amplitude values may not be bounded. For example, a large amplitude control parameter can significantly increase the blur amount in large-motion scenes, resulting in unnatural large-motion blur images. Such unbounded amplitude values may not be optimal for amplitude augmentation. To address this, we bound the absolute amplitude values adjusted by $\alpha$ to the values between $1.0\times$ and $1.5\times$, and $\alpha$ is randomly chosen to keep the bounded range. As a result, the bounded amplitude augmentation policy achieves the best performance. In summary, we randomly select $\alpha$ to keep the absolute amplitude values between $1.0\times$ and $1.5\times$ for amplitude augmentation, and we use random $\beta$ values between -90° and 90° for phase augmentation unless otherwise specified.

Table 7: Effects on amplitude / phase augmentation policies. The best results are indicated in bold.

| Method | Adjustment | Range | PSNR ↑ | SSIM ↑ |
|---|---|---|---|---|
| GeoSyn (ours) | Amplitude $\alpha$ | $0.5\times \sim 1.0\times$ | 32.68 | 0.932 |
| | | $1.0\times \sim 1.5\times$ | 32.80 | **0.933** |
| | | $1.0\times \sim 2.0\times$ | 32.78 | **0.933** |
| | | $1.0\times \sim 4.0\times$ | 32.76 | **0.933** |
| | | Bounded Amplitudes | **32.82** | **0.933** |
| | Phase $\beta$ | $-15° \sim 15°$ | 32.73 | 0.932 |
| | | $-30° \sim 30°$ | **32.78** | **0.933** |
| | | $-45° \sim 45°$ | **32.78** | **0.933** |
| | | $-90° \sim 90°$ | **32.78** | **0.933** |
| | | $-180° \sim 180°$ | 32.65 | 0.932 |

**Data augmentation during the deblurring training.** When training the deblurring model, we use both real blur image $B$ and augmented blur image $B_{\text{aug}}$ based on the accumulated gradient (AG) strategy. Specifically, it relies on accumulating gradients in both real data and synthesized data (1:1) and then backpropagation. As it seems similar to $2\times$ increases of mini-batch size, we experiment on $2\times$ increases of mini-batch size using AG technique, resulting in slight performance improvement, $32.50 \rightarrow 32.56$ dB, as shown in "Accumulated gradient" of Table 8. We clarify that the performance improvement is not due to gradient accumulation itself, but rather our data augmentation scheme $32.50 \rightarrow 32.99$ dB as shown in Table 8.

Table 8: Effects on accumulated gradients. The best results are indicated in bold.

| Methods | PSNR ↑ | SSIM ↑ |
|---|---|---|
| None | 32.50 | 0.928 |
| Accumulated gradient | 32.56 | 0.929 |
| GeoSyn (ours) | **32.99** | **0.936** |

## C    CONSIDERATION ON THE COMPENSATION NETWORK

As shown in Fig. 2, we use the compensation network to address the photometric issues. Basically, RealBlur (Rim et al., 2020) and RSBlur (Rim et al., 2022) datasets are acquired using dual-camera systems. These systems consist of two cameras with different lenses or sensors, resulting in color drifts. Furthermore, since blur and sharp images require different exposure time, this leads to variations in brightness and contrast. Although such photometric inconsistencies are corrected by post-processing, they are not fully compensated. This means that the synthesized blur image using a sharp image may be different from a ground-truth blur image in terms of color, brightness, or contrast. This difference may disrupt motion modeling. To confirm that the compensation network is necessary, we conduct experiments with the configurations: (1) training our blur synthesizer without the compensation network and using the estimated blur image $\tilde{B}$ for blur data augmentation, (2) training our blur synthesizer with the compensation network and using the estimated blur image $\tilde{B}$ for blur data augmentation (ours), (3) training our blur synthesizer with the compensation network and using the compensated blur image $h_\xi(\tilde{B})$ for blur data augmentation, and (4) training our blur synthesizer with the compensation network and using the compensated blur image $h_\xi * (\tilde{B})$ with finetuning. For simplicity, we refer to configuration (1) as C1, configuration (2) as C2, config-

uration (3) as C3, and configuration (4) as C4. As shown in Table 9, the results show that the C2 gives the best deblurring performance (32.99 dB). We believe that the usage of the compensation network during the synthesizer training enables the 3D-aware vector fields to focus exclusively on learning blur components, thereby free from photometric variations such as color drifts and sensor differences. On the other hand, the C1 shows a sub-optimal result (32.86 dB) since it may suffer from the photometric issues for motion modeling. Furthermore, when training deblurring models with our blur data augmentation scheme, various blur patterns can be generated by manipulating the amplitude and phase of the vector field. Such modified blur patterns may be unseen to the compensation network, which may cause unexpected artifacts or blur effects after the compensation network. Therefore, the C3 leads to sub-optimal performance (32.78 dB), compared to that of the C2 (32.99 dB). To address this, we finetune the compensation network during the deblurring training, which is indicated as the C4. We observe that it somewhat handles unseen blur patterns by finetuning the compensation model ($32.78 \rightarrow 32.89$ dB), but it lags behind our configuration, C2. As a result, we empirically prove that the C2 gives the best performance, and use it unless otherwise specified.

Table 9: Effects on the compensation network. $h_\xi*$ is a finetuned version of $h_\xi$. The best results are indicated in bold.

| Configurations | Compensation Net | Synthetic Blur | PSNR ↑ | SSIM ↑ |
|---|---|---|---|---|
| C1 | | $\tilde{B}$ | 32.86 | 0.935 |
| C2 (ours) | ✓ | $\tilde{B}$ | **32.99** | **0.936** |
| C3 | ✓ | $h_\xi(\tilde{B})$ | 32.78 | 0.934 |
| C4 | ✓ | $h_\xi*(\tilde{B})$ | 32.89 | 0.934 |

## D    COMPARISON RESULTS WITH ID-BLAU

We compare our method with ID-Blau (Wu et al., 2024) to demonstrate its effectiveness. We conduct experiments across various network architectures such as MIMO-UNet+ (Cho et al., 2021) and FFT-former (Kong et al., 2023), and datasets such as GoPro (Nah et al., 2017) and RealBlur-J (Rim et al., 2020). As shown in Table. 10, our GeoSyn gives better performance than ID-Blau. In particular, our GeoSyn achieves a PSNR of 33.01 dB on GoPro, outperforming 32.93 dB obtained by ID-Blau in MIMO-UNet+. We believe that our blur synthesizer effectively accounts for explicit 3D motion modeling, leading to better performance. Furthermore, our method shows remarkable performance improvement on RealBlur-J, compared with that of ID-Blau. Notably, while ID-Blau shows the performance improvement in FFTformer from 32.62 to 32.88 dB, our GeoSyn achieves a significant performance improvement, reaching 33.68 dB. Unlike ID-Blau which requires video frame images, our GeoSyn is compatible with training on RealBlur-J which only contains blur-sharp image pairs. Hence, our method can generate more dataset-specific motion patterns for data augmentation, such that it yields better subsequent deblurring performance on RealBlur-J.

Table 10: Comparison results against ID-Blau. The best results are indicated in bold.

| Methods | GoPro | | RealBlur-J | |
|---|---|---|---|---|
| | PSNR↑ | SSIM↑ | PSNR↑ | SSIM↑ |
| MIMO-UNet+ | 32.44 | 0.957 | 31.92 | 0.916 |
| + ID-Blau | 32.93 | 0.961 | 31.96 | 0.921 |
| + GeoSyn | **33.01** | **0.962** | **32.55** | **0.925** |
| FFTformer | 34.21 | 0.969 | 32.62 | 0.932 |
| + ID-Blau | 34.36 | **0.970** | 32.88 | 0.934 |
| + GeoSyn | **34.39** | **0.970** | **33.68** | **0.938** |

## E    GENERALIZATION ABILITY

To demonstrate the generalization ability of our method, we conduct cross-dataset evaluations. First, we train both our blur synthesis and deblurring models on RealBlur (i.e., GeoSyn-R) and test on RealBlur (Rim et al., 2020), RSBlur (Rim et al., 2022) and BSD (Zhong et al., 2020). Additionally, we train our blur synthesis model on GoPro and deblurring model on RealBlur (i.e., GeoSyn-G) and test again on RealBlur, RSBlur and BSD. As shown in Table 11, our GeoSyn-R shows remarkable

performance on RealBlur, and our GeoSyn-G demonstrates promising generalization performance on RSBlur and BSD. This relies on what dataset is used for training our blur synthesizer. Specifically, the GeoSyn-R is trained on RealBlur, enabling it to generate diverse and dataset-specific blur patterns that contribute to performance improvement on RealBlur. Even though the GeoSyn-R uses only RealBlur in both trainings, it also shows good generalization results on RSBlur and BSD because it can generate numerous and diverse blur patterns during the deblurring training. On the other hand, our GeoSyn-G leverages separate datasets for the blur synthesizer (GoPro) and the deblurring model (RealBlur), enabling it to benefit from multiple blur datasets. As a result, it achieves superior generalization performance (see the results on BSD Test set).

Table 11: Ablation study on the cross-data validation. The best results are indicated in bold.

| Train (Deblur) | Test | Train (Synthesizer) | Methods | NAFNet | | MIMO-UNet+ | |
|---|---|---|---|---|---|---|---|
| | | | | PSNR | SSIM | PSNR | SSIM |
| RealBlur | RealBlur | - | No Aug | 32.50 | 0.928 | 31.92 | 0.919 |
| | | GoPro | ID-Blau | 32.70 | 0.932 | 31.96 | 0.921 |
| | | RealBlur | GeoSyn-R | **32.99** | **0.936** | **32.55** | **0.925** |
| | | GoPro | GeoSyn-G | 32.94 | 0.935 | 32.47 | 0.924 |
| RealBlur | RSBlur | - | No Aug | 30.61 | 0.809 | 29.72 | 0.790 |
| | | GoPro | ID-Blau | 30.90 | 0.814 | 29.43 | 0.786 |
| | | RealBlur | GeoSyn-R | 30.91 | 0.815 | **29.95** | **0.794** |
| | | GoPro | GeoSyn-G | **30.98** | **0.815** | 29.81 | 0.794 |
| RealBlur | BSD | - | No Aug | 29.67 | 0.893 | 29.25 | 0.891 |
| | | GoPro | ID-Blau | 30.42 | 0.907 | 28.93 | 0.882 |
| | | RealBlur | GeoSyn-R | 30.83 | 0.911 | 29.73 | 0.899 |
| | | GoPro | GeoSyn-G | **30.91** | **0.912** | **29.91** | **0.904** |

## F COMPREHENSIVE ANALYSIS ON GEOSYN

### F.1 EFFECTS ON $\lambda$

We examine the hyperparameters $\lambda_1$ and $\lambda_2$ to confirm performance sensitivity across the hyperparameters. We train our blur synthesizers using combinations of $\lambda_1 = \{0.1, 1.0\}$, $\lambda_2 = \{0.1, 1.0\}$, resulting in the pairs $(\lambda_1, \lambda_2)$ such as $(0.1, 0.1)$, $(0.1, 1.0)$, $(1.0, 0.1)$, and $(1.0, 1.0)$. The results are given in Table. 12. We observe that the higher impact ($\lambda_2 = 1.0$) of invertible geometric loss gives better performance. This means that it accounts for the importance of geometric consistency for blur synthesis. In contrast, we observe that the greater impact ($\lambda_1 = 1.0$) of Laplacian smoothing loss leads to decreased deblurring performance. This is because a stronger smoothing constraint impedes the learning of diverse motion patterns, reducing data diversity for data augmentation. Subsequently, it limits performance improvements.

Table 12: Ablation study on hyperparameters $\lambda_1$ and $\lambda_2$. The best results are indicated in bold.

| Methods | $\lambda_1$ | $\lambda_2$ | PSNR ↑ | SSIM ↑ |
|---|---|---|---|---|
| GeoSyn (ours) | 0.1 | 0.1 | 32.90 | 0.935 |
| | 0.1 | 1.0 | **32.99** | **0.936** |
| | 1.0 | 0.1 | 32.76 | 0.933 |
| | 1.0 | 1.0 | 32.84 | 0.934 |

### F.2 BLUR DIVERSITY AND ITS RELATIONSHIP TO $M$

When the number of camera positions $M$ is small (e.g., $M = 4$), the expressive power of complex motion using only four camera exposures becomes inherently constrained. Namely, complex motion trajectory is represented by a simplified form, leading to a lack of blur diversity. In contrast, increasing $M$ to 16 provides the capability to capture intricate motion patterns, leading to better blur diversity. As shown in Fig. 9, we observe that the larger number of $M$, i.e., $M = 16$ yields more accurate motion results, ultimately promoting a wider range of blur patterns. Therefore, it leads to better final deblurring results as discussed in Section 4.5.

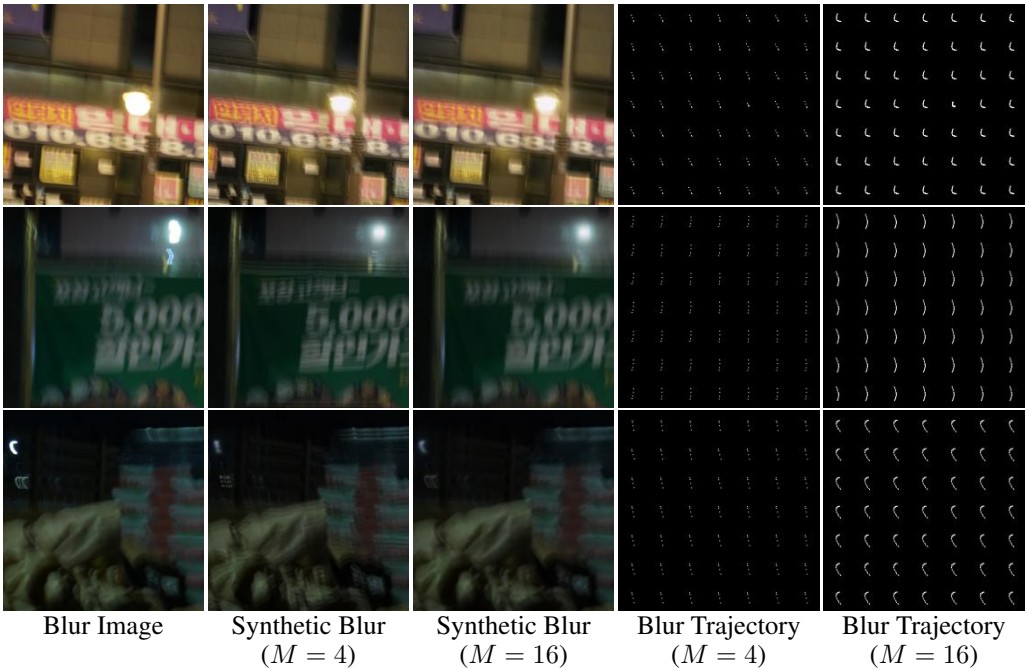

| Blur Image | Synthetic Blur $(M = 4)$ | Synthetic Blur $(M = 16)$ | Blur Trajectory $(M = 4)$ | Blur Trajectory $(M = 16)$ |

Figure 9: Blur diversity and its relationship with the number of camera positions $M$.

### F.3 PERFORMANCE SENSITIVITY AND ITS RELATIONSHIP TO GEOMETRIC CONSISTENCY

To investigate the performance sensitivity to the accuracy of the vector field, we experiment by introducing random perturbations to the vector fields. We believe that the robustness of the final deblurring performance is closely tied to the geometric consistency of the vector field. As shown in Table 13, without geometric consistency ($\lambda_2 = 0.0$), the vector field lacks geometric coherence, and thus even no perturbation leads to performance degradation (32.52 dB) compared to the baseline performance, i.e., no data augmentation (32.56 dB). In contrast, with strong geometric consistency ($\lambda_2 = 1.0$), the deblurring model remains robust to the perturbations of the vector field, consistently outperforming the baseline (indicated in bold). For mid-level geometric consistency ($\lambda_2 = 0.5$), the vector field is generally robust to small perturbations (indicated in bold), but large perturbations disrupt its geometric coherence and adversely affect performance (32.51 dB). These observations highlight the critical role of our geometric consistency regularization in mitigating the performance sensitivity to the accuracy of the vector field. Namely, blur data augmentation with inaccurate vector fields reduces performance gain but does not degrade baseline deblurring performance, as long as our motion estimator is trained under strong geometric consistency.

Table 13: Ablation study on performance sensitivity under perturbations to the vector fields. The performance exceeding the baseline is indicated in bold.

| Methods | $\lambda_2$ | Perturbations | | | |
|---|---|---|---|---|---|
| | | 0 | 0.001 | 0.005 | 0.01 |
| Baseline (No Augmentation) | - | 32.56 | | | |
| With GeoSyn | 1.0 | **32.92** | **32.90** | **32.83** | **32.70** |
| | 0.5 | **32.77** | **32.77** | **32.75** | 32.51 |
| | 0.0 | 32.52 | 32.51 | 32.49 | 32.49 |

### F.4 VISUAL COMPARISON ON BLUR TRAJECTORIES

Our primal goal is to build a controllable blur synthesizer that estimates motions from a single blur image. This enables our blur synthesizer to be directly applicable to various blur datasets (blur-sharp image pairs) such as GoPro (Nah et al., 2017), RealBlur (Rim et al., 2020), RSBlur (Rim et al., 2022), BSD (Zhong et al., 2020), and ReLoBlur (Li et al., 2023a). However, we can compare

blur trajectories estimated by our blur synthesizer with those obtained from video frames using an off-the-shelf optical flow model, e.g., RAFT (Teed & Deng, 2020). To this end, we utilize GoPro, which provides video frame images that allow us to extract optical flow maps. Then, we convert these optical flow maps into vector fields to represent a real-like blur trajectory, which provides a straightforward way to verify that our blur trajectory aligns well with real ones. The results are visualized in Fig. 10. We found that the blur trajectories are nearly identical, confirming the alignment between our blur trajectories and the real one. Despite estimating blur trajectories from single blur images, our method is comparable to those derived from video frames.

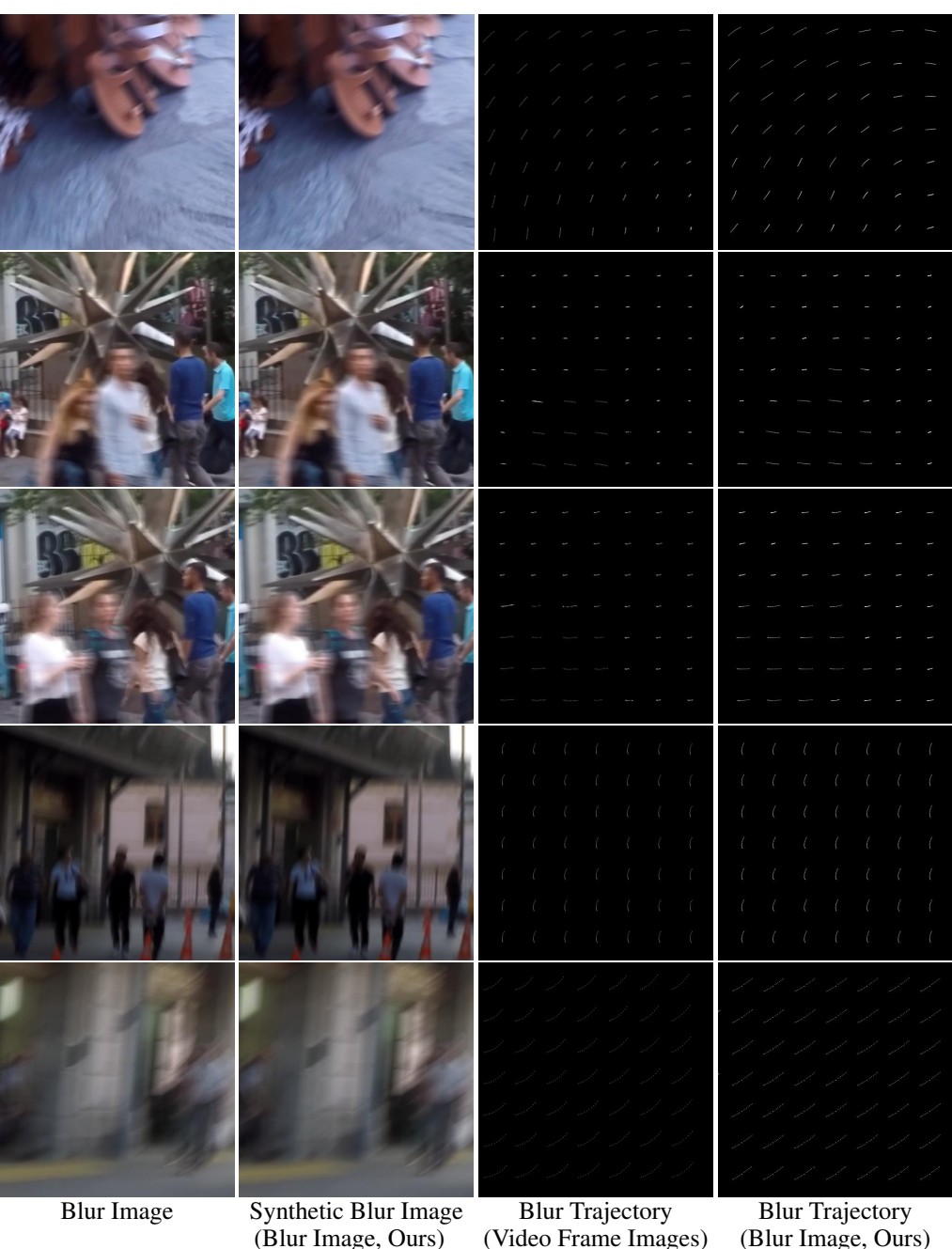

| Blur Image | Synthetic Blur Image (Blur Image, Ours) | Blur Trajectory (Video Frame Images) | Blur Trajectory (Blur Image, Ours) |

Figure 10: Comparison results on blur trajectories (video frame images vs. single blur image).

# G   VECTOR FIELD GENERATION IN PYTORCH PSEUDOCODE

```python
def vec2skew(v):
    zero = torch.zeros(1, dtype=torch.float32, device=v.device)
    skew_v0 = torch.cat([ zero,    -v[2:3],   v[1:2]])  # (3, 1)
    skew_v1 = torch.cat([ v[2:3],   zero,    -v[0:1]])
    skew_v2 = torch.cat([-v[1:2],   v[0:1],   zero])
    skew_v = torch.stack([skew_v0, skew_v1, skew_v2], dim=0)  # (3, 3)
    return skew_v  # (3, 3)

def Exp(r):
    skew_r = vec2skew(r)  # (3, 3)
    norm_r = r.norm() + 1e-15
    eye = torch.eye(3, dtype=torch.float32, device=r.device)
    R = eye + (torch.sin(norm_r) / norm_r) * skew_r + \
        ((1 - torch.cos(norm_r)) / norm_r**2) * (skew_r @ skew_r)
    return R

def make_c2w(r, t):
    R = Exp(r)  # (3, 3)
    c2w = torch.cat([R, t.unsqueeze(1)], dim=1)  # (3, 4)
    return c2w

def complex_to_polar(z):
    z_complex = z[..., 0] + 1j * z[..., 1]
    return torch.abs(z_complex), torch.angle(z_complex)

def polar_to_complex(amplitude, phase):
    real = amplitude * torch.cos(phase)
    imag = amplitude * torch.sin(phase)
    return torch.stack((real, imag), axis=-1)

def vec_field_gen(motion_results, control_params, img_size):
    # 2D Rigid --> r, t = [b, num_poses * 3]
    # 3D-Aware Residual Field --> amp_local, phase_local = [b, h, w]
    r, t, amp_local, phase_local = motion_results

    # Control parameters are randomly chosen for blur data augmentation
    amp_control, phase_control = control_params   # (1, 0) for training

    r_split, t_split = r.split(3, dim=1), t.split(3, dim=1)
    total_grid, total_grid_invert  = [], []
    for i in range(len(r_split)):
        rigid_2d = []
        for j in range(len(r_split[i])):
            rigid_2d.append(make_c2w(r_split[i][j], t_split[i][j]))
        rigid_2d = torch.stack(rigid_2d)

        grid_2d_cano = F.affine_grid(torch.eye(3, 3).cuda().unsqueeze(0).
            repeat(rigid_2d.size()[0], 1, 1)[:, :2, :3], img_size)
        grid_2d_rigid = F.affine_grid(rigid_2d[:, :2, :3], img_size)

        res_grid_2d_rigid = grid_2d_rigid - grid_2d_cano[..., :2]

        amp_global, phase_global = complex_to_polar(res_grid_2d_rigid)
        res_grid_3d_aware = polar_to_complex(amp_global * amp_local *
            amp_control, phase_global + phase_local + phase_control)

        total_grid.append(res_grid_3d_aware + grid_2d_cano)
        total_grid_invert.append(-res_grid_3d_aware + grid_2d_cano)
    total_grid = torch.stack(total_grid)
    total_grid_invert = torch.stack(total_grid_invert)

    return total_grid, total_grid_invert
```

## H    BLUR SYNTHESIS AND DATA AUGMENTATION IN PYTORCH PSEUDOCODE

```python
model_motion_est, compensation_net = models      # neural networks
def blur_synthesis(blur_img, sharp_img, control_params=(1,0)):
    motion_results = model_motion_est(blur_img)
    b,c,h,w = bchw = sharp_img.size()
    grids,grids_inv = vec_field_gen(motion_results, control_params, bchw)

    num_poses, out, out_invert, lap_reg_loss = 16, [], [], 0
    for i in range(num_poses):
        grid, grid_inv = grids[i], grids_inv[i]
        lap_reg_loss = lap_reg_loss + compute_lap_loss(grid)

        gaussian_noise = torch.randn_like(sharp_img).cuda() * 0.0112
        view_img_syn = torch.clamp(torch.nn.functional.grid_sample(
            sharp_img, grid, mode='bilinear') + gaussian_noise, 0, 1)

        out.append(view_img_syn)
        out_invert.append(torch.nn.functional.grid_sample(view_img_syn,
            grid_inv, mode='bilinear'))

    out = torch.stack(out)
    out_avg = out.mean(0)

    out_invert = torch.stack(out_invert)      # [num_poses,b,c,h,w]
    out_invert = out_invert.reshape(num_poses*b,c,h,w)
    sharp_img_reshaped = sharp_img.unsqueeze(0).repeat(num_poses,1,1,1,1)
        .reshape(num_poses*b,c,h,w)
    geo_reg_loss = nn.L1Loss()(out_invert, sharp_img_reshaped)

    compen_pred_blur_img = compensation_net(out_avg)
    blur_syn_results = torch.cat((compen_pred_blur_img,out_avg),1)

    return blur_syn_results, lap_reg_loss, geo_reg_loss

def blur_data_augmentation(blur_img, target_img, blur_synthesis):
    B = blur_img.size()[0]
    amp_control = torch.empty(B,1,1).uniform_(1.0, 2.0)
    phase_control = torch.empty(B,1,1).uniform_(-1.0,1.0) * (math.pi/2.0)
    control_params = (amp_control, phase_control)

    perm_idx = torch.randperm(B)
    target_shuf = target_img[perm_idx]
    pred_imgs,_,_ = blur_synthesis(blur_img, target_shuf, control_params)

    return pred_imgs[:,3:,:,:].detach()

def compute_lap_loss(grid):
    grid = (torch.clamp(grid, -1.0, 1.0) + 1.0) / 2.0
    center = grid[:, 1:-1, 1:-1,:]
    up, down = grid[:, :-2, 1:-1,:], grid[:, 2:, 1:-1,:]
    left, right = grid[:, 1:-1, :-2,:], grid[:, 1:-1, 2:,:]
    laplacian = 4 * center - (up + down + left + right)

    return ((laplacian ** 2).sum(dim=[1, 2, 3])).mean()

def total_loss(blur_img, sharp_img, blur_synthesis):
    pred_imgs, loss_lap, loss_geo = blur_synthesis(blur_img, sharp_img)

    pred_blur, pred_blur_avg = pred_imgs[:,:3,:,:], pred_imgs[:,3:,:,:]
    loss_blur1 = nn.L1Loss()(pred_blur, blur_img)
    loss_blur2 = nn.L1Loss()(pred_blur_avg, blur_img)

    return loss_blur1 + loss_blur2 + 1.0*loss_geo + 0.1*loss_lap
```

# I GENERALIZATION TO REAL-WORLD BLUR IMAGES

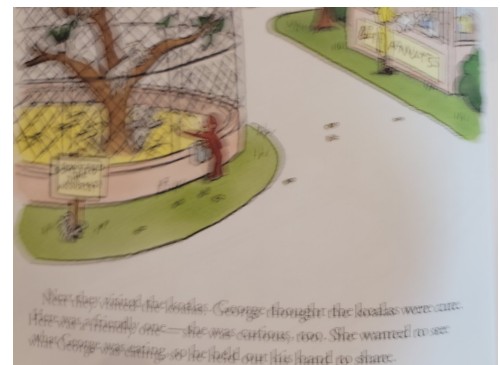

Blur Input

FFTformer

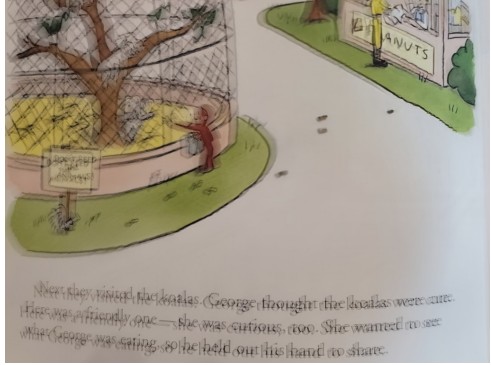
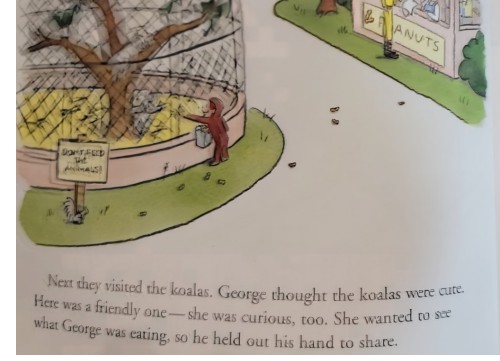

FFTformer + ID-Blau

FFTformer + GeoSyn (ours)

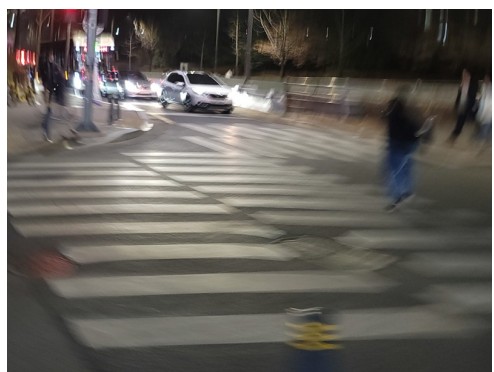
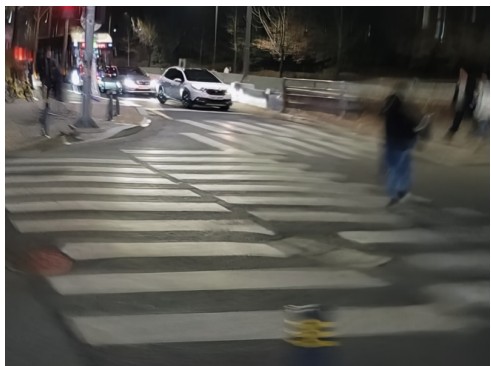

Blur Input

FFTformer

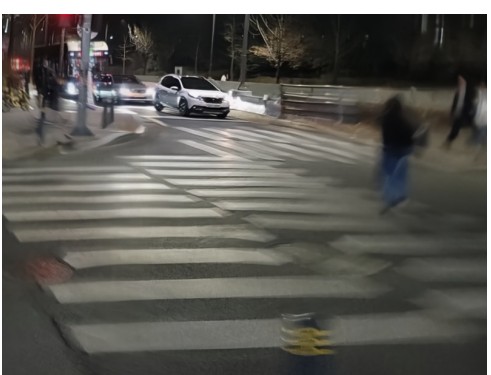
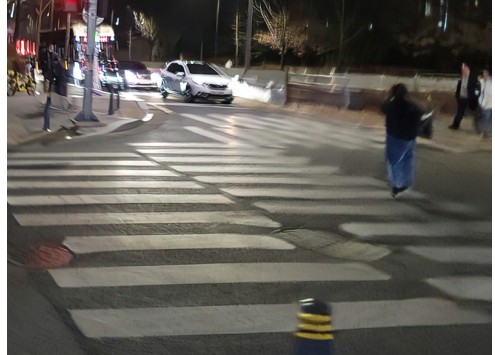

FFTformer + ID-Blau

FFTformer + GeoSyn (ours)

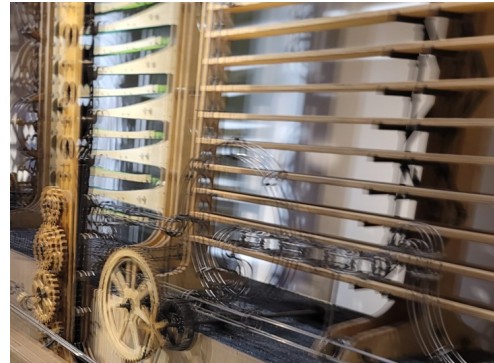
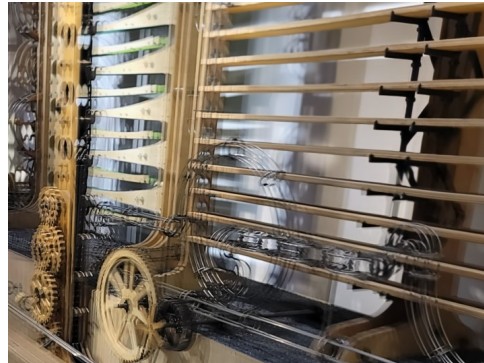

Blur Input                                    FFTformer

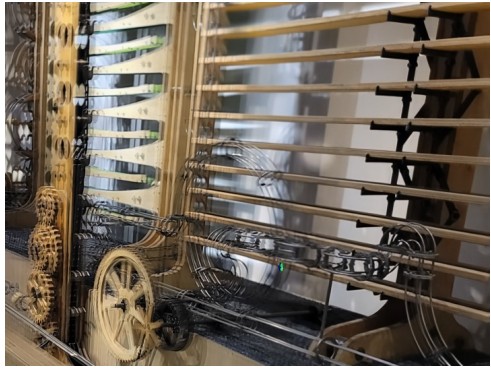
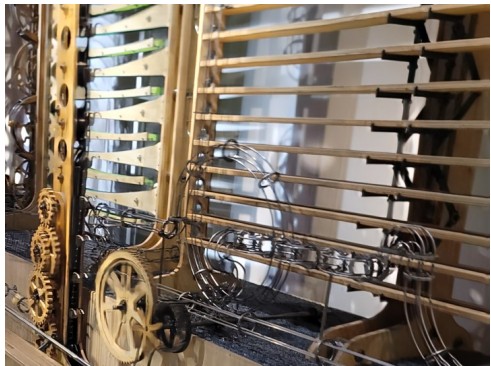

FFTformer + ID-Blau                    FFTformer + GeoSyn (ours)

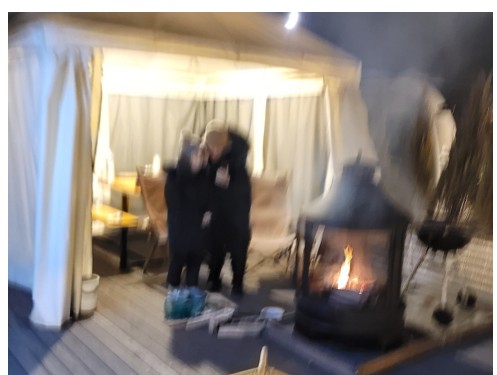
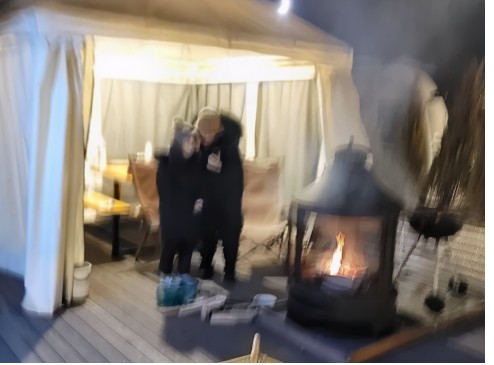

Blur Input                                    FFTformer

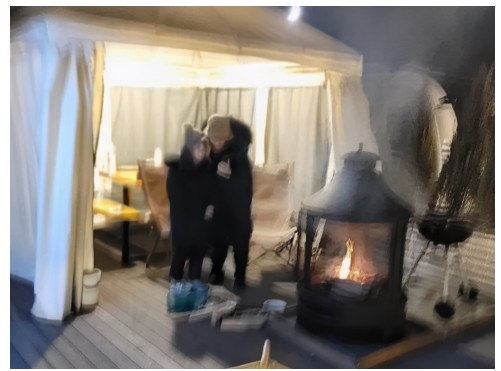
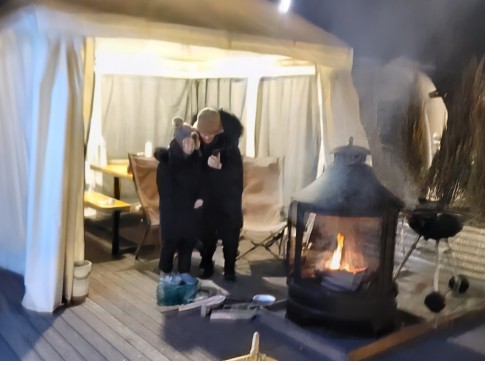

FFTformer + ID-Blau                    FFTformer + GeoSyn (ours)

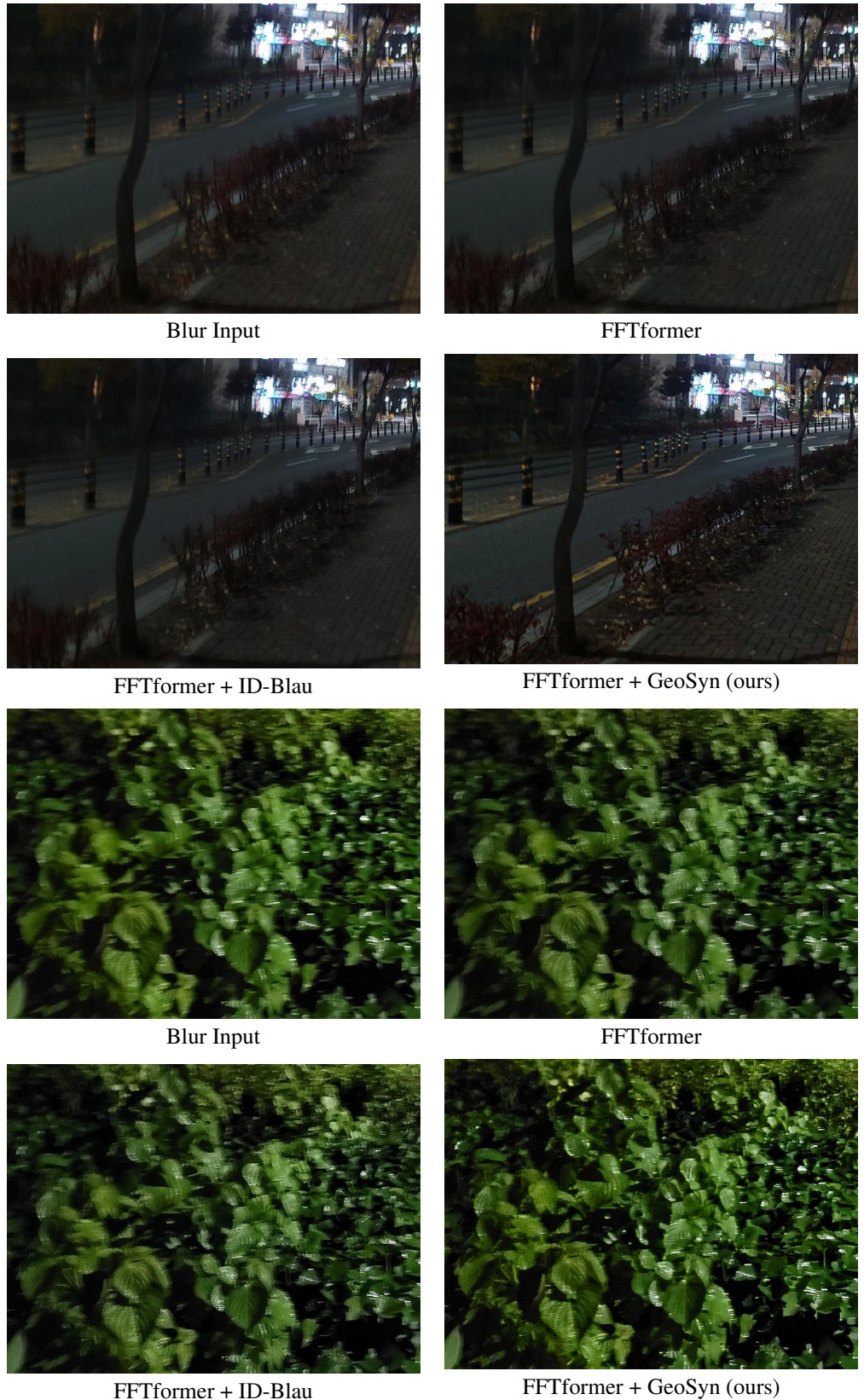

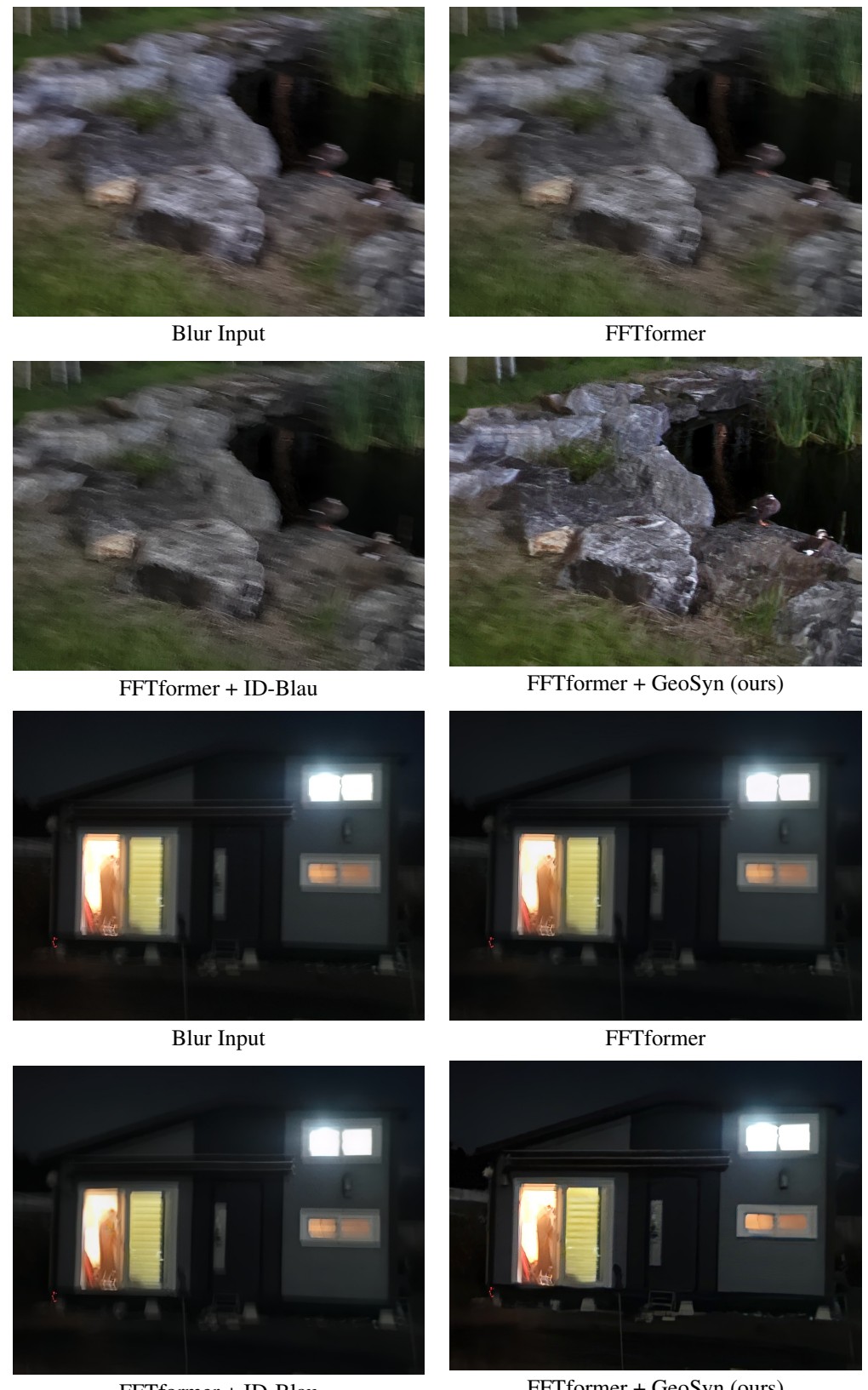

Figure 11: Visual results on real-world blur images.

## J    QUALITATIVE RESULTS ON GOPRO

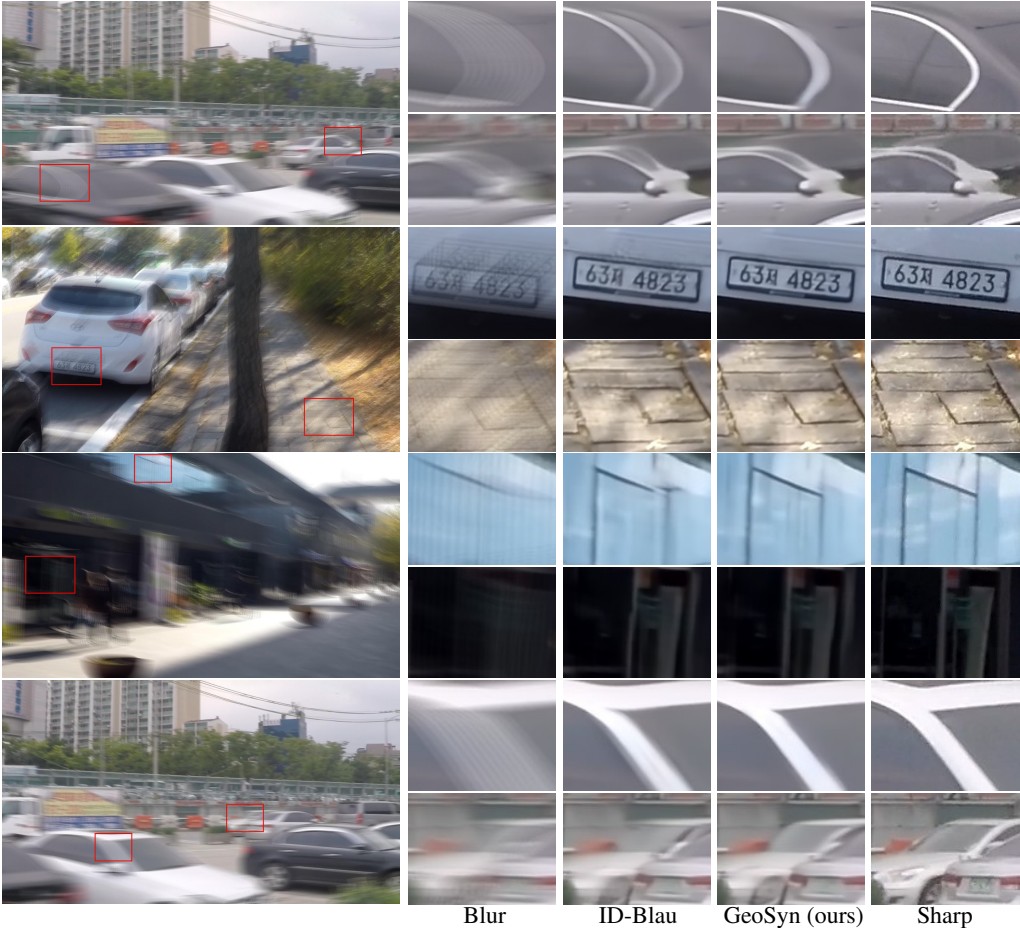

        Blur       ID-Blau   GeoSyn (ours)   Sharp

Figure 12: Qualitative results on GoPro (Nah et al., 2017). All models are trained with GoPro and are based on MIMO-UNet (Cho et al., 2021).

## K   QUALITATIVE RESULTS ON REALBLUR-J

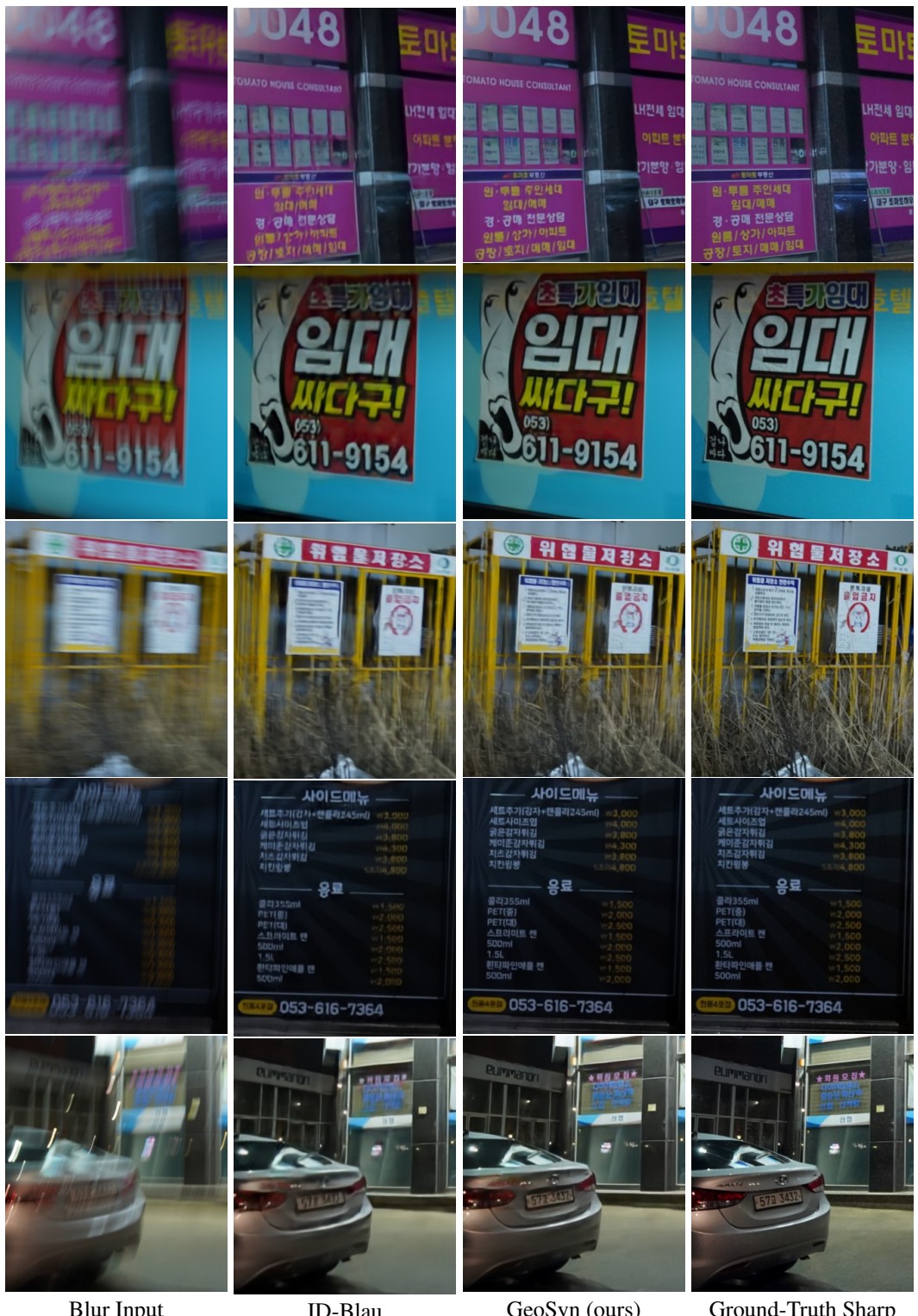

| Blur Input | ID-Blau | GeoSyn (ours) | Ground-Truth Sharp |

Figure 13: Qualitative deblurring results on RealBlur-J (Rim et al., 2020). All models are trained with RealBlur-J and are based on NAFNet-64 (Chen et al., 2022).

## L BLUR SYNTHESIS AND TRAJECTORY RESULTS

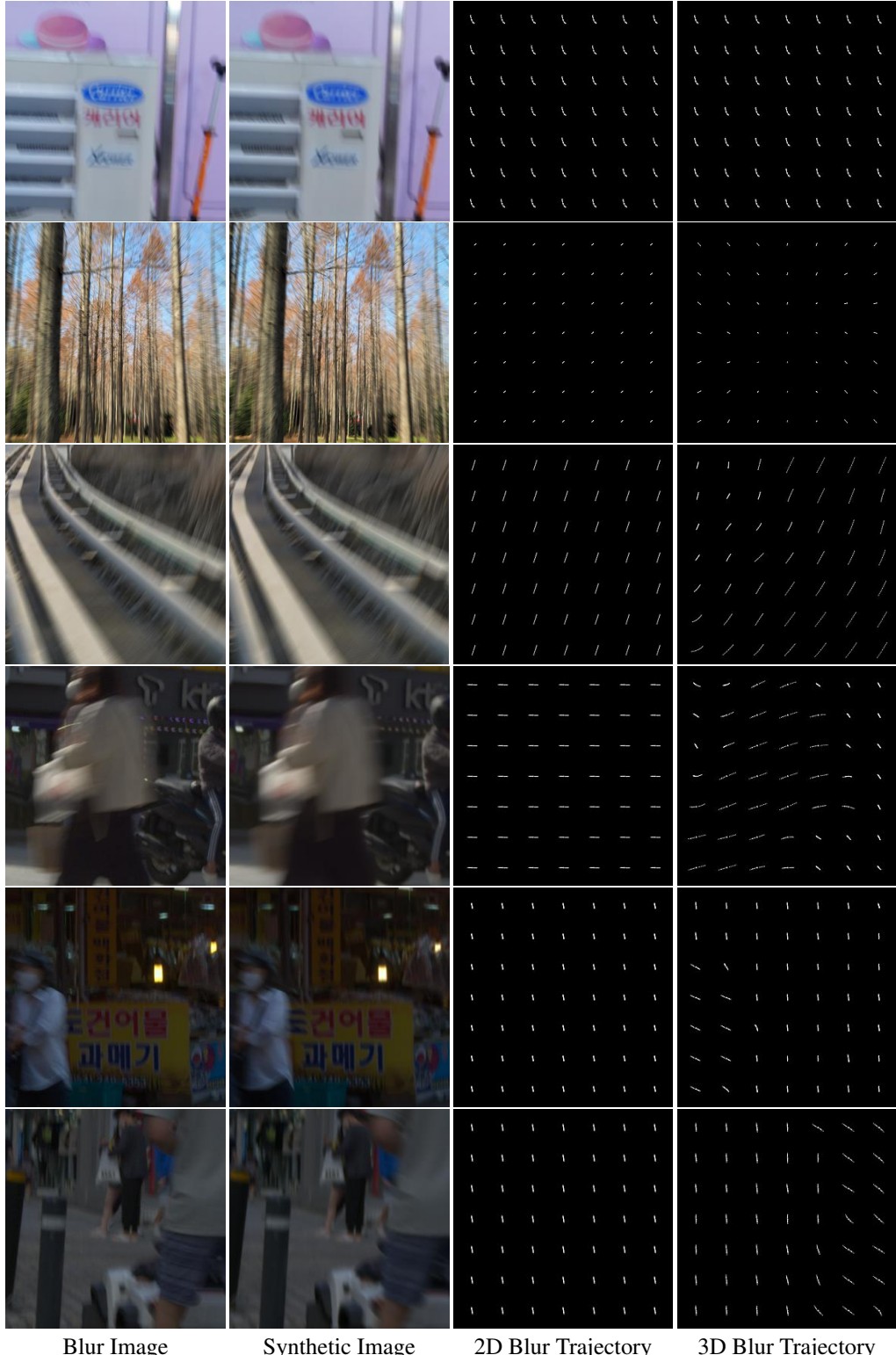

| Blur Image | Synthetic Image | 2D Blur Trajectory | 3D Blur Trajectory |

Figure 14: Synthesized blur images and blur trajectories on 2D camera motion (1st row), 3D camera motion (2 - 3rd rows), and 3D object + camera motion examples (4 - 6th rows).

## M    BLUR SYNTHESIS AND TRAJECTORY RESULTS ON OBJECT MOTION

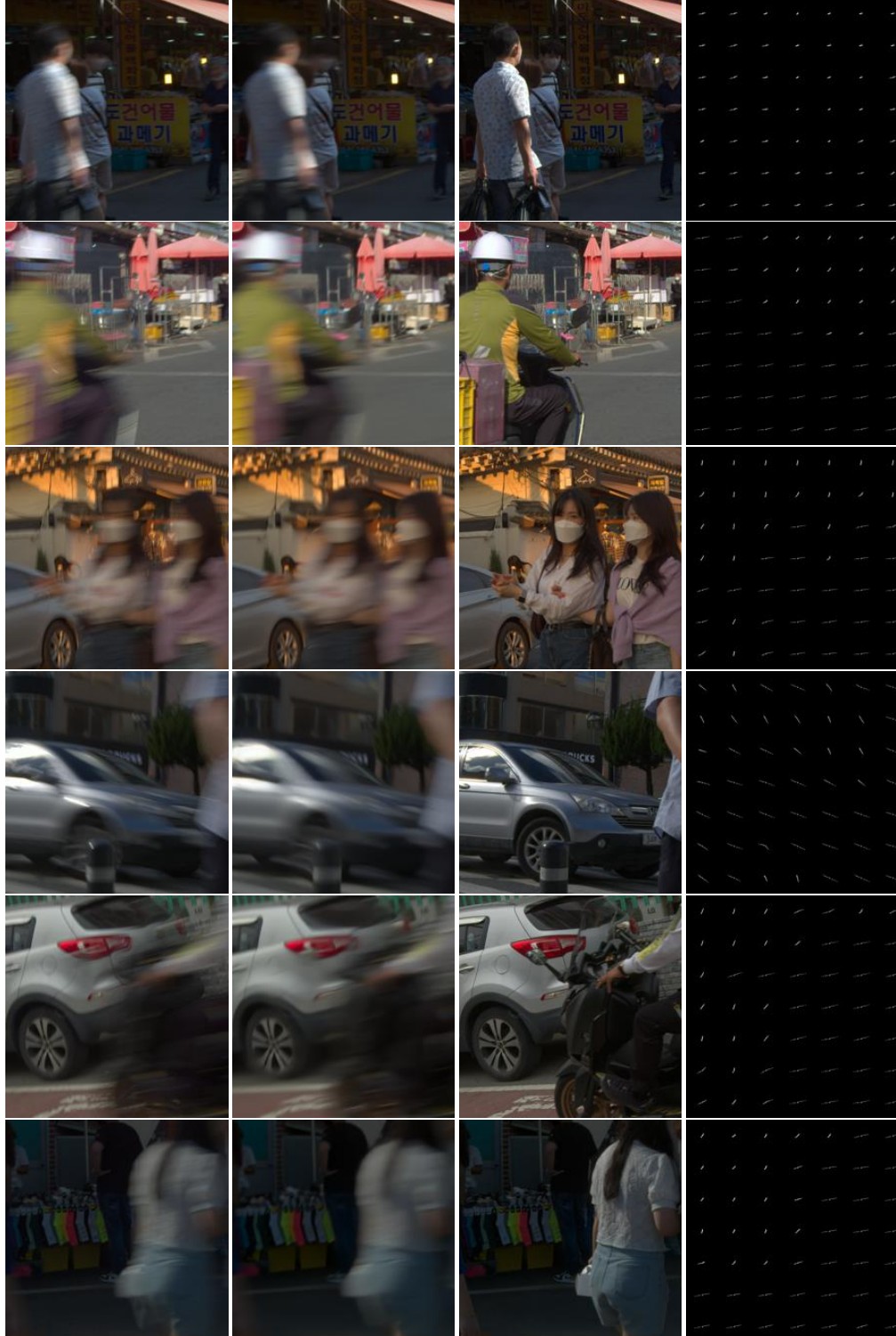

| Blur Image | Synthetic Blur Image | Sharp Image | Blur Trajectory |

Figure 15: Blur synthesis results and their blur trajectories for object motion using RSBlur dataset (Rim et al., 2022).

## N    CONTROLLABLE BLUR IMAGE SYNTHESIS ON GOPRO

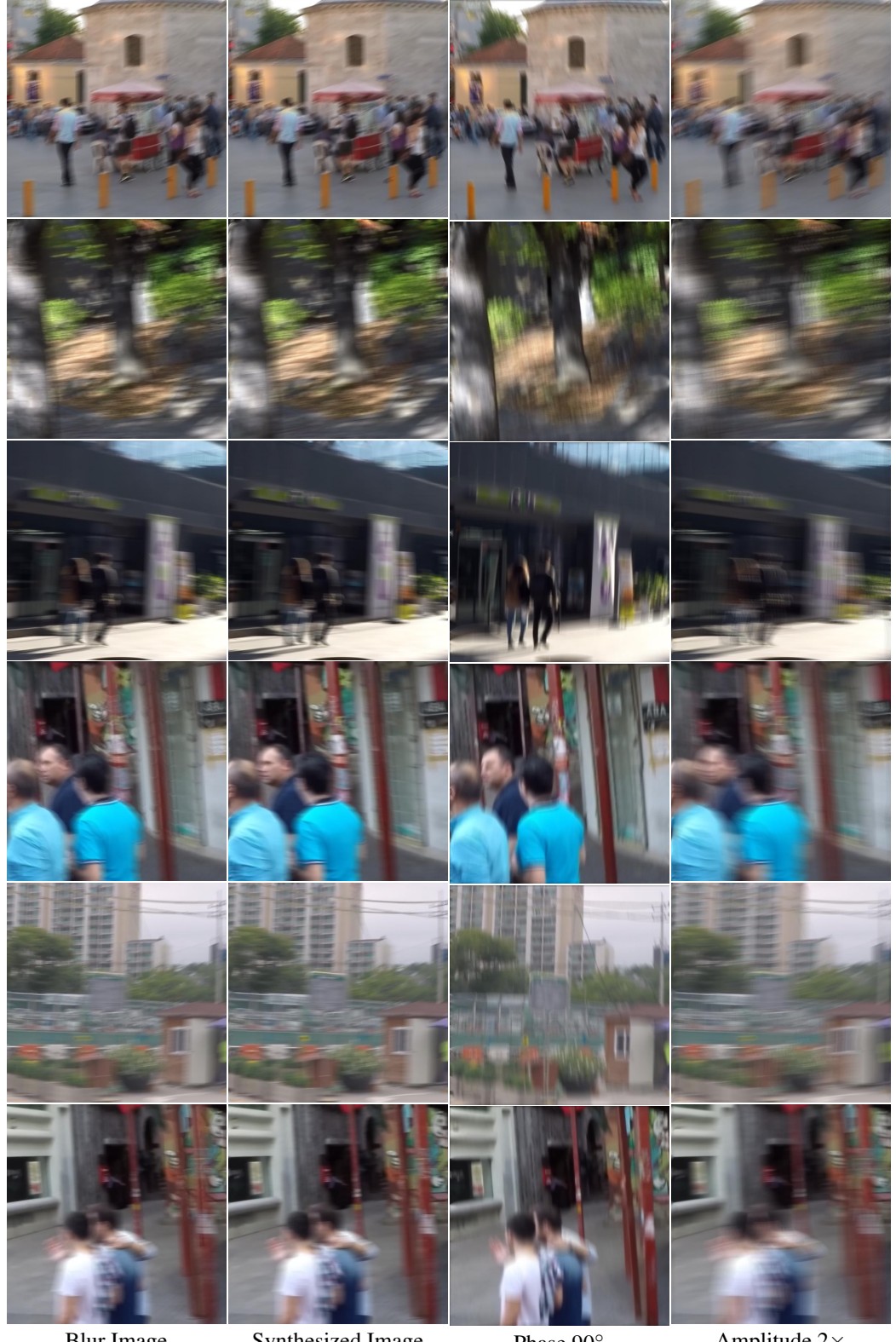

| Blur Image | Synthesized Image | Phase 90° | Amplitude 2× |

Figure 16: Controllable blur image synthesis on GoPro (Nah et al., 2017).

## O    CONTROLLABLE BLUR IMAGE SYNTHESIS ON REALBLUR-J

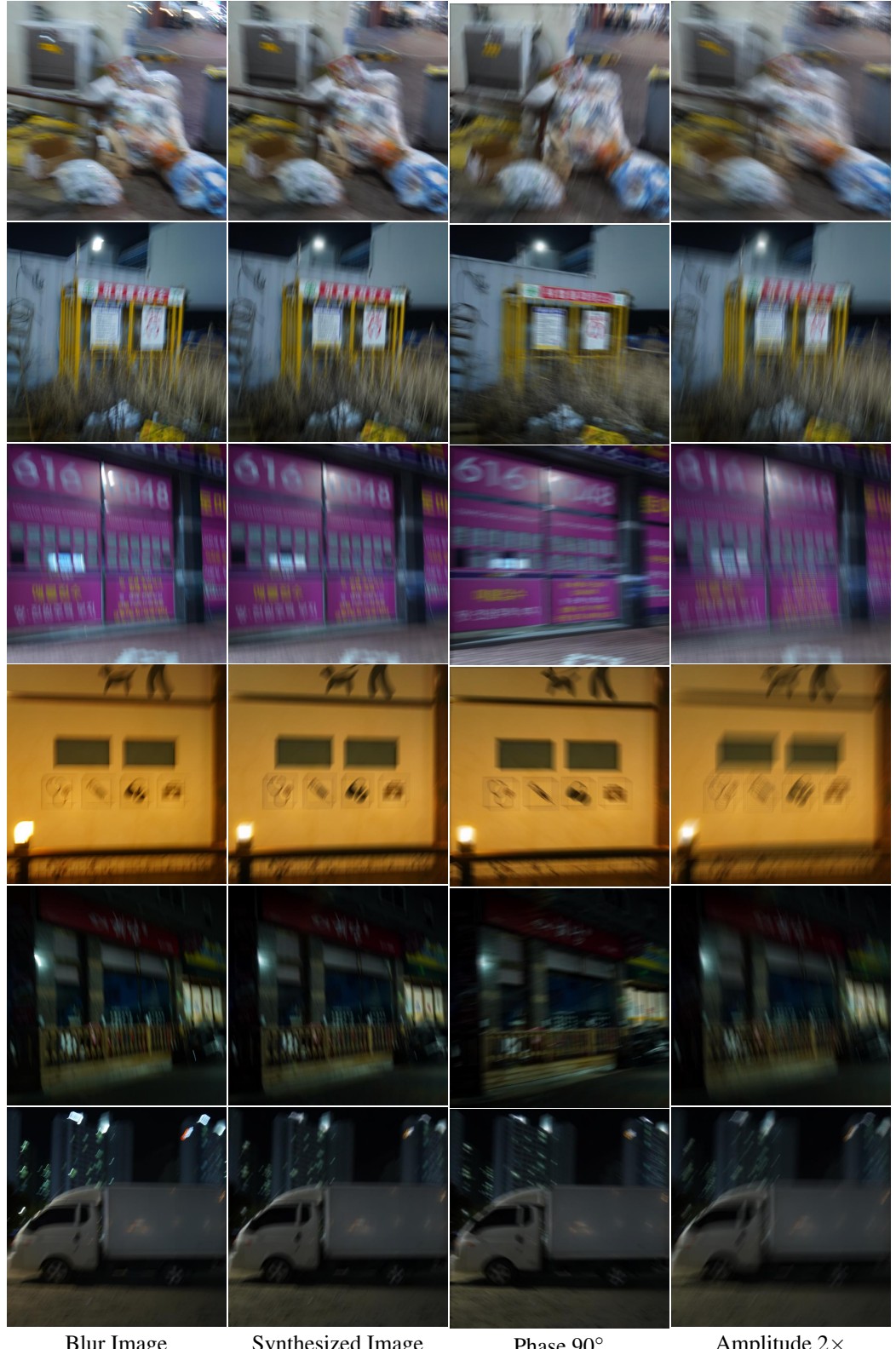

| Blur Image | Synthesized Image | Phase 90° | Amplitude 2× |

Figure 17: Controllable blur image synthesis on RealBlur-J (Rim et al., 2020).

