# OpenReview forum: "Controllable Blur Data Augmentation Using 3D-Aware Motion Estimation"
_ICLR.cc/2025/Conference — ICLR 2025 Poster_

### Official Review · Reviewer_iVrH · 2024-10-22

**Soundness:** 2
**Presentation:** 3
**Contribution:** 2
**Rating:** 6
**Confidence:** 5

**Summary:**

This paper presents a novel, controllable blur data augmentation method aimed at enhancing the performance of learning-based deblurring models. The authors propose using a neural network to learn both a parametric vector field (2D rigid transformation) and a non-parametric vector field (3D residual field) to simulate 3D transformations. By adjusting these learned vector fields, the method can generate diverse and realistic blur patterns for data augmentation.

**Strengths:**

The motivation is well-founded: improving deblurring performance through data augmentation is highly cost-effective.
The paper is clearly written and easy to follow.
Based on the experimental results, the proposed method achieves state-of-the-art performance in deblurring tasks.

**Weaknesses:**

The upper performance limit for video deblurring is typically higher than for image deblurring, as videos provide more temporal information. It remains unclear whether the proposed data augmentation method can handle video sequences while maintaining temporal consistency, which is crucial for real-world applications like Apple Live Photos.

The paper repeatedly emphasizes the term "physical," which this reviewer finds somewhat exaggerated. From a single image, a neural network theoretically cannot disentangle object motion from camera motion due to the numerous possible motion combinations. At best, the method can adapt to the motion patterns of the dataset it is trained on. Additionally, even if the transformation accurately models 3D motion, the transformed image is still 2D and inherently lacks occluded information, making it less "physical" than synthesis methods using nearby video frames.

The paper does not sufficiently address blur caused by object motion (This reviewer did not see any blurred images dominated by object motion). This raises doubts about the method's "physical" nature and its effectiveness in real-world scenarios involving complex object motion.

> "ID-Blau requires ground-truth video frames to extract optical flows, so it cannot be trained on datasets containing only blurred and sharp images, such as RealBlur and RSBlur."

Given the advantage of synthesis methods using nearby frames, for a fair comparison, ID-Blau could utilize datasets like BSD[1] and RBI[2], which contain blur-sharp video pairs. Specifically, the BSD[1] dataset offers subsets with varying blur intensities. A cross-dataset evaluation between RealBlur and BSD is necessary to demonstrate the generalization capability of the proposed data augmentation method.

[1] Efficient spatio-temporal recurrent neural network for video deblurring. In *Computer Vision–ECCV 2020: 16th European Conference, Glasgow, UK, August 23–28, 2020, Proceedings, Part VI 16* (pp. 191-207). Springer International Publishing.

[2] Blur interpolation transformer for real-world motion from blur. In *Proceedings of the IEEE/CVF Conference on Computer Vision and Pattern Recognition* (pp. 5713-5723).

**Questions:**

Please address the concerns raised in the weaknesses section.

---

> ### Author Response · Authors · 2024-11-22
> **Official Comment by Authors (1/2)**
>
> Dear Reviewer iVrH,
>
> We sincerely thank the reviewer for your valuable comments and efforts. The insights have helped us clarify our contributions. We have revised our draft (highlighted in blue) in response to the insightful and valuable comments. We do our best to clarify each of the concerns carefully as follows:
>
> ----
> **[W1] Handle video sequences by our method.**
>
> We agree that video deblurring generally achieves higher performance than image deblurring due to the additional temporal information available. However, we wish to clarify that our primal goal is to build a controllable blur synthesizer that estimates motions from a single blur image. This enables our blur synthesizer to be directly applicable to various blur datasets (blur-sharp image pairs) such as GoPro [1], RealBlur [2], RSBlur [3], BSD [4], and ReLoBlur [5]. Therefore, extending our blur synthesizer to handle video sequences is beyond the scope of our work.
>
> Nevertheless, to address the reviewer's concern, we compare blur trajectories estimated by our blur synthesizer with those obtained from video frames using an off-the-shelf optical flow model, e.g., RAFT [6]. Specifically, we extract the optical flow maps from the video frame images in GoPro [1] and convert the optical flow maps into vector fields to represent a real-like blur trajectory. The results are visualized in the revised draft. Please refer to Fig. 10 of the Appendix. We observe that the blur trajectories are nearly identical, confirming the alignment between our blur trajectories and the real ones.
> Despite estimating blur trajectories from single blur images, our method is comparable to those derived from video frames.
>
> ----
> **[W2] Emphasizing the term "physical" and handling occluded regions.**
>
> We agree that the term "physical" may cause confusion.
> Our intention was to emphasize the incorporation of 3D motion modeling, which accounts for more realistic blur patterns, as opposed to the other methods that estimate 2D non-uniform kernels.
> Furthermore, as discussed in Section 3.1, our method explicitly follows a camera exposure model to approximate the blurring process. Therefore, we used the term "physical" for these reasons and do not imply that our blur synthesizer is inherently more physically grounded than the synthesizer using video frames.
> We have revised the misleading term "physical" to "realistic" and "3D aspects" throughout our draft.
>
> As our blur synthesizer relies on a single target image to generate a synthesized blur image, handling occluded regions within the target image remains a challenge. While the occlusion issue may arise in certain cases, we believe that our 3D-aware blur synthesizer handles the majority of motion patterns encountered in real-world scenarios.
> Despite using only a single blurred image, our GeoSyn demonstrates better generalization ability compared to ID-Blau [7] which leverages video frame images. Please see the Common Response 1 for detailed generalization results.
>
> ----
> **[W3] Blur synthesis results on object motion.**
>
> To address the reviewer's concerns, we provide some blur synthesis results on object motion using RSBlur [3] in the revised draft. Please refer to Fig. 17 of the Appendix.
>
> ----
> **[W4] Limited motion patterns and cross-dataset evaluation.**
>
> We appreciate the reviewer for highlighting this point. To respond to the statement "At best, the method can adapt to the motion patterns of the dataset it is trained on.", we would like to clarify that our method can leverage separate blur datasets for the blur synthesizer and the deblurring model. We chose to use the same dataset for both training to ensure a fair comparison with other methods, addressing any concerns about unfair comparisons arising from the use of additional datasets.
> Even though we use the same dataset in both training, our blur synthesizer can improve model generalization. Specifically, if a blur dataset contains $3,000$ blur images and the deblurring model is trained over $1,000$ epochs, it generates approximately $3,000,000$ different synthetic blur images by randomly varying blur intensities, directions, and scenes during the deblurring training. This extensive diversity enhances the generalization ability of the deblurring model, enabling it to handle a wide range of unseen blur scenarios.
> As suggested by the reviewer, we conducted a cross-dataset evaluation to demonstrate the generalization ability of our blur synthesizer. Please refer to Common Response $1$ for detailed generalization results.

---

> ### Author Response · Authors · 2024-11-22
> **Official Comment by Authors (2/2)**
>
> **References**
>
> [1] Nah et al., Deep multi-scale convolutional neural network for dynamic scene deblurring, CVPR 2017
>
> [2] Rim et al., Real-world blur dataset for learning and benchmarking deblurring algorithms, ECCV 2020
>
> [3] Rim et al., Realistic blur synthesis for learning image deblurring, ECCV 2022
>
> [4] Zhong et al., Efficient spatio-temporal recurrent neural network for video deblurring, ECCV 2020
>
> [5] Li et al., Real-World Deep Local Motion Deblurring, AAAI 2023

---

> > ### Comment · Reviewer_iVrH · 2024-11-24
> >
> > Thanks for the authors' effort in addressing the reviewers’ concerns. While some issues have been resolved, certain aspects still require further clarification and improvement:
> >
> > Regarding Fig. 10 in the Appendix: Was the synthesizer trained and tested on the GoPro dataset? How does the model perform when trained on RealBlur and tested on GoPro? Please clarify.
> >
> > In Fig. 17, this reviewer observes that the control over object-caused blur appears very coarse, as the background is also affected. In addition, the intention behind the comment, “The paper does not sufficiently address blur caused by object motion (This reviewer did not see any blurred images dominated by object motion).” was to request the restored results specifically for cases of object-caused blur (real-world examples are preferred).

---

> > > ### Author Response · Authors · 2024-11-25
> > > **Official Comment by Authors**
> > >
> > > Dear Reviewer iVrH,
> > >
> > > Thank you for the additional comments and for providing us with the opportunity to further clarification of our method.
> > >
> > > ----
> > > **How does the model perform when trained on RealBlur and tested on GoPro?**
> > >
> > > The previous results presented in Fig. 10 are based on training and testing with the GoPro dataset. We have included the blur trajectory results from the model trained with RealBlur in Fig. 10 of the revised draft. As shown in Fig. 10, our blur synthesizer trained with the RealBlur dataset performs well in some examples such as natural camera motions, but struggles with rotational motion and object motion within the GoPro dataset. Here, we would like to emphasize that this result is due to dataset-specific variations and does not affect the significance of our contributions.
> > > We believe the motion patterns of RealBlur significantly differ from those of GoPro. Specifically, RealBlur primarily captures camera motion, while GoPro includes both camera motion and object motion. Furthermore, the motion patterns in GoPro are tailored for video recording, whereas RealBlur is specialized to capture natural hand motions.
> > > Given these fundamental differences, the blur synthesizer trained with RealBlur would result in suboptimal performance when evaluated on GoPro due to the inherent discrepancy in motion patterns between the two blur datasets.
> > >
> > > ----
> > > **Background motion effect.**
> > >
> > > We believe that the background motion effect of some examples in Fig. 17 arises from the regression-to-the-mean effect, which is a common challenge in image restoration tasks. This effect leads to a tendency to learn an averaged motion distribution across the overall scene, including local object motion regions and background. As a result, it may impact static backgrounds, resulting in an overestimation or underestimation of its motions.
> > > To address this, one potential solution could be to leverage object blur masks to focus more on object blur regions, which helps reduce unintended adjustments on the background. We leave it for future work.
> > >
> > > ----
> > > **Deblurred results on real-world object motion blur images.**
> > >
> > > The ReLoBlur dataset [1] is a blur dataset for real-world object motions. It is acquired using a beam splitter, which provides sharp reference images. Therefore, it facilitate accurate and reliable comparisons. To ensure a fair comparison under the ReLoBlur dataset, we initially attempted to test it using models trained with the GoPro dataset, including ID-Blau and GeoSyn. However, the GoPro-trained models leads to significant artifacts due to the discrepancy between synthetic (e.g., GoPro) and realistic datasets (e.g., ReLoBlur). Therefore, we instead utilize models trained with RSBlur [2] including NAFNet-64 and NAFNet-64 + GeoSyn to evaluate the ReLoBlur test set. Since ID-Blau does not provide the RSBlur-trained model, it is excluded from the comparison. We have provided multiple examples in the revised draft. Please refer to Fig. 18 of the Appendix.
> > >
> > > ----
> > > **References**
> > >
> > > [1] Li et al., Real-World Deep Local Motion Deblurring, AAAI 2023
> > >
> > > [2] Rim et al., Realistic blur synthesis for learning image deblurring, ECCV 2022

---

> > > ### Author Response · Authors · 2024-11-26
> > > **Official Comment by Authors**
> > >
> > > Dear Reviewer iVrH,
> > >
> > > We initially provided some examples on ReLoBlur dataset to demonstrate the deblurring performance for object motions. However, as suggested by the reviewer, we have expanded our evaluation to include real-world blur images with object motion which are captured by us. Note that we only captured the dynamic scenes for moving vehicles due to privacy issues. We have updated our draft to include the deblurring results on real-world scenarios. Please see Fig. 19 of the Appendix.
> > >
> > > Thank you very much!
> > >
> > > Authors

---

> > > > ### Comment · Reviewer_iVrH · 2024-11-27
> > > >
> > > > Although the generalization of the proposed methodology is still a concern, after discussion and additional experiments, this reviewer considered it above average and decided to increase the score.

---

> > > > > ### Author Response · Authors · 2024-11-28
> > > > > **Official Comment by Authors**
> > > > >
> > > > > Dear Reviewer iVrH,
> > > > >
> > > > > Thank you for raising the score! The valuable and constructive feedback has helped us clarify our contribution. Thank you once again for taking the time to review our paper and the opportunity to refine our work. Please feel free to reach out with any additional questions or suggestions.
> > > > >
> > > > > Best regards,
> > > > >
> > > > > Authors

---

### Official Review · Reviewer_RsRS · 2024-10-31

**Soundness:** 4
**Presentation:** 4
**Contribution:** 3
**Rating:** 8
**Confidence:** 4

**Summary:**

This paper introduces a method for controllable blur image synthesis to address the challenges in capturing datasets for deblurring (GeoSyn). Specifically, it estimates M-many 3D vector fields for each pixel of a blur image and reconstructs the original blur image through projections of these estimated vector fields. During evaluation, blur with various trajectories can be generated by adjusting these 3D-aware vector fields. The method combines both parametric and non-parametric trajectory fields to capture both object motion and 3D camera motion. Training deblurring models with this augmentation leads to improved performance.

**Strengths:**

1. The paper effectively identifies key challenges in blur data augmentation and addresses them logically and directly by leveraging 3D spaces.

2. The writing and figures clearly illustrate the motivation and methodology of the proposed approach.

3. The ablation studies and experiments provide strong evidence of the proposed augmentation's effectiveness.

**Weaknesses:**

1. Limitation of motion trajectories: Since motion trajectories are determined independently of content, unrealistic trajectories can occur. For instance, part of a car might move left while another part moves right. Additionally, the method cannot represent motion caused by shape-changing objects. While real-world blur includes cases (e.g., folding fingers or blinking eyes), the proposed method only addresses part of this blur complexity.

2. The proposed method increases computational costs. The reviewer acknowledges that the augmentation scheme could help build an efficient deblurring mode, as mentioned in lines 507-513 of the main paper. However, could the authors provide quantitative metrics (latency, GMACs, and memory usage) when adopting GeoSyn?

**Questions:**

1. Could the authors provide the distribution of 2D-based motion trajectories (obtained via optical flow between the sharp and original blur images) and the distribution of the proposed 3D-aware vector fields? The reviewer would like to see if the augmented blur trajectories align well with real ones. This could be visualized using t-SNE projection of the motion vectors.

2. How do the authors obtain the camera intrinsics, as this could be ill-posed from a single image? Do the authors fix the intrinsic parameters and then optimize only the extrinsic scaling factor?

3. For the synthetic dataset, the reverted sharp image mean({\sum^{M}_{*=1}}\tilde{S}_{τ}_{*}) should be identical to the sharp image, based on Equation 10. Could the authors provide quantitative results on this?

---

> ### Author Response · Authors · 2024-11-22
> **Official Comment by Authors**
>
> Dear Reviewer RsRS,
>
> We sincerely thank the reviewer for your valuable comments and efforts. The insights have helped us clarify our contributions. We have revised our draft (highlighted in blue) in response to the insightful and valuable comments. We do our best to clarify each of the concerns carefully as follows:
>
> ----
> **[W1] Limitation of motion trajectories.**
>
> Thank you for pointing this out.
> We agree that our blur trajectories are content-independent and occasionally lead to perceptually implausible blur results when the blur trajectories are applied to other scene images. Regarding non-rigid motion cases, our blur synthesizer does not fully represent non-rigid motion blur caused by shape changes (e.g., blinking eyes or folding fingers), as such motions cannot be adequately rendered by using the grid sampling based on
> a single target image.
> While these issues may arise in certain cases, we believe that our 3D-aware blur synthesizer handles the majority of motion patterns encountered in real-world scenarios. Despite using only a single blurred image, our GeoSyn demonstrates better generalization ability compared to ID-Blau [1] which leverages video frame images. Please see the Common Response 1 for detailed generalization results.
>
> ----
> **[W2] Computational costs.**
>
> We would like to clarify that our GeoSyn is only used for training. Therefore, it does not increase the computational cost in the evaluation stage. We have revised Section 4.1 of our draft to clarify this. To discuss it further, we provide the computational costs in the training phase, such as GMACs, model size, and training latency.
> The GMACs is calculated for the image size of $256 \times 256$ and the training latency is measured for one iteration with the batch size $32$ with V100 8 GPUs.
> As shown in the table below, utilizing GeoSyn with the network architecture of NAFNet-64 corresponds to an increase of $1.13$ GMACs, $1.0$ MB in model size, and $0.024$ seconds in training latency.
>
>
> | **Method**     | **GMACs**         | **Model Size (MB)**       | **Training Latency (s)**        |
> |-----------------|-------------------|-----------------------|------------------------|
> | NAFNet-64       | 63.50            | 63.64                | 0.172                 |
> | + GeoSyn        | 64.63 (+1.13)    | 64.64 (+1.0)         | 0.196 (+0.024)        |
>
>
> ----
> **[Q1] Visual comparison on blur trajectories.**
>
> Thank you for the insightful comments. To address this, we utilize GoPro dataset [2], which provides video frame images that allow us to extract optical flow maps. Then, we convert these optical flow maps into vector fields to represent a real-like blur trajectory, which allows us to verify that our blur trajectory aligns well with real one. The results are visualized in the revised draft. Please refer to Fig. 10 of the Appendix. We observe that the blur trajectories are nearly identical, confirming the alignment between our blur trajectories and the real ones.
> Despite estimating blur trajectories from single blur images, our method is comparable to those derived from video frames.
>
> ----
> **[Q2] How to obtain the camera intrinsics?**
>
> The intrinsic parameters are not explicitly estimated. Instead, they are inherently embedded within the projected 3D residual vector, which is directly optimized by the neural network to estimate 3D motions. We discuss the relationship between the camera intrinsic and projected 3D residual vector in the revised draft. Please refer to Section A of the Appendix for further details.
>
> ----
> **[Q3] Quantitative comparison of reverted sharp and ground-truth sharp images.**
>
> Thank you for the feedback. We measure the PSNR and SSIM between the average of the reverted sharp images and the ground-truth sharp images in RealBlur-J. The results are PSNR of $41.07$ dB and SSIM of $0.9902$.
>
> ----
> **References**
>
> [1] Wu et al., Id-blau: Image deblurring by implicit diffusion-based reblurring augmentation, CVPR 2024
>
> [2] Nah et al., Deep multi-scale convolutional neural network for dynamic scene deblurring, CVPR 2017

---

> > ### Comment · Reviewer_RsRS · 2024-11-25
> >
> > Thank you to the authors for their efforts in addressing the reviewers' concerns. Incorporating an additional segmentation mask (like *) to address rigid-body motion blur could be a potential improvement in the final version; however, as it is not officially published yet, this remains a minor suggestion. The reviewer will maintain the previous score.
> >
> > *GS-Blur: A 3D Scene-Based Dataset for Realistic Image Deblurring, NeurIPS 2024.

---

> > > ### Author Response · Authors · 2024-11-25
> > > **Official Comment by Authors**
> > >
> > > Dear Reviewer RsRS,
> > >
> > > We are glad to hear that our responses have addressed the reviewer's concerns. The valuable and constructive feedback has helped us clarify our contribution. We also appreciate the suggestion regarding the potential use of an additional segmentation mask for rigid-body motion blur. We will carefully consider it for further improvements. Thank you once again for the thoughtful review and the opportunity to refine our work.
> > >
> > > Best regards,
> > >
> > > Authors

---

> ### Author Response · Authors · 2024-11-25
> **Gentle Reminder**
>
> Dear Reviewer RsRS,
>
> Thank you again for your time and efforts in reviewing our paper.
>
> As the discussion period draws close, we kindly remind you that two days remain for further comments or questions. We would appreciate the opportunity to address any additional concerns you may have before the discussion phase ends.
>
> Thank you very much!
>
> Authors

---

### Official Review · Reviewer_BF8H · 2024-11-04

**Soundness:** 2
**Presentation:** 3
**Contribution:** 2
**Rating:** 6
**Confidence:** 4

**Summary:**

The manuscript presents a 3D-aware blur synthesizer designed to generate diverse blur images for data augmentation, which further improves the deblurring performance of existing approaches.  The blur synthesizer enables controllable blur data augmentation not requiring any extra data during the training by combining 2D motion and 3D residual component.

**Strengths:**

The strengths of this paper could be summarized as follows:

1. Comprehensive experiments have been conducted to demonstrate the
effectiveness of the proposed method.


2. The proposed blur image augmentation method could be beneficial to relative research.

**Weaknesses:**

There are some concerns and problems that should be further clarified and addressed:
1. The amplitude-phase integration directly offers controllability to motion patterns behind the blur images. However, it seems unable to explicitly adjust the overall blur intensity. Have the authors ever considered providing a way to control it? Such as adjusting exposure time or introducing a blur intensity parameter.


2. Please give more details about how to derive the projected 3D residual vector.


3. Authors have explored the effects of different values of M on the final deblurring performance. But what concerns me most is the correlation between the value of M and blur diversity, which could possibly provide more controllability to the blur synthesizer. For example, the visualizations or metrics showing how blur patterns change as M varies, or specific experiments to show blur diversity and its relationship to M.


4. As observed by the authors, inaccurate depth measurements can lead to performance degradation. On the other hand, Motion estimation itself is a very complex problem, even under 2D space. So, how can the authors guarantee that the inaccuracy of the estimated displacement field will not corrupt the final deblurring performance? Or authors can prove that deblurring effects are not sensitive to the accuracy of the motion field. For example, when training the blur synthesizer, we can randomly apply perturbations to the generated displacement fields to see whether it would make a difference.


5. In the experiments, the blur synthesis model is separately trained for different datasets. Nevertheless, this reduces the credit of the proposed model having better generalization ability compared with existing approaches. I would like to see more experiments involving a blur model trained on a single dataset and tested on different datasets.

6. Please provide more comparisons of real-world blurred images captured by the authors and clarify whether the compared models utilize existing blur augmentation methods.

**Questions:**

See the Weaknesses above.

---

> ### Author Response · Authors · 2024-11-22
> **Official Comment by Authors (1/2)**
>
> Dear Reviewer BF8H,
>
> We sincerely thank the reviewer for your valuable comments and efforts. The insights have helped us clarify our contributions. We have revised our draft (highlighted in blue) in response to the insightful and valuable comments. We do our best to clarify each of the concerns carefully as follows:
>
> ----
> **[W1] Regarding the adjustment of overall blur intensity.**
>
> As discussed in Section 3.4, we introduce an amplitude control parameter $\alpha$ to control the overall blur intensity. Note that the same value of $\alpha$ is applied to all vector fields generated from a single blur image to adjust the overall blur intensity. Additionally, the value of $\alpha$ is randomly determined for each blur image.
> This allows us to simulate variations in blur intensity, e.g., exposure time, for the same motion trajectory. For example, when $\alpha$ is set to $2$, the amplitude of the overall vector field results in doubling the blur intensity. Meanwhile, scaling it by 0.5 reduces the overall blur intensity by half. We have already provided the results of these adjustments as shown in Fig. 1, Fig. 15, and Fig. 16. Also, we have revised Section 3.4 of our draft to provide further clarification.
>
> ----
> **[W2] How to derive the projected 3D residual vector.**
>
> For further details on deriving the projected 3D residual vector, please refer to Section A of Appendix in the revised draft.
>
> ----
> **[W3] Blur diversity and its relationship to M.**
>
> Thank you for the insightful comment. When $M$ is small (e.g., $M=4$), the expressive power of complex motion using only four camera exposures becomes inherently constrained. Namely, complex motion trajectory is represented by a simplified form, leading to a lack of blur diversity. In contrast, increasing $M$ to $16$
> provides the capability to capture intricate motion patterns, leading to better blur diversity. We have provided multiple examples in the revised draft. Please refer to Fig. 9 of the Appendix. We observe that the larger number of $M$, i.e., $M=16$ yields more accurate motion results, ultimately promoting a wider range of blur patterns. Therefore, it leads to better final deblurring results as discussed in Section 4.5.
>
> ----
> **[W4] Performance sensitivity and its relationship to vector fields.**
>
> We appreciate the reviewer’s insightful comment regarding the correlation between deblurring performance and the estimated vector field. To address this concern, we conducted the suggested experiment by introducing random perturbations to the vector fields. We believe that the robustness of the final deblurring performance is closely tied to the geometric consistency of the vector field. As shown in the table below, without geometric consistency ($\lambda_2=0.0$), the vector field lacks geometric coherence, and thus even no perturbation leads to performance degradation ($32.52$ dB) compared to the baseline performance, i.e., no data augmentation ($32.56$ dB). In contrast, with strong geometric consistency ($\lambda_2=1.0$), the deblurring model remains robust to the perturbations of the vector field, consistently outperforming the baseline (indicated in bold).
> For mid-level geometric consistency ($\lambda_2=0.5$), the vector field is generally robust to small perturbations (indicated in bold), but large perturbations disrupt its geometric coherence and adversely affect performance ($32.51$ dB). These observations highlight the critical role of our geometric consistency regularization in mitigating the performance sensitivity to the accuracy of the vector field.
> In other words, blur data augmentation with inaccurate vector fields reduces performance gain but does not corrupt the final deblurring performance, e.g., baseline deblurring performance, as long as our motion estimator is trained under strong geometric consistency.
>
>
> | Methods                      | λ₂   |  Perturbation |     Perturbation |     Perturbation  |Perturbation|
> |------------------------------|-------|---------------|-------|-------|-------|
> |                                    |        | 0             | 0.001 | 0.005 | 0.01  |
> | Baseline (No Augmentation)   | -     | 32.56   |
> | With GeoSyn                  | 1.0   | **32.92**     | **32.90** | **32.83** | **32.70** |
> |                              | 0.5   | **32.77**     | **32.77** | **32.75** | 32.51 |
> |                              | 0.0   | 32.52         | 32.51  | 32.49  | 32.49 |

---

> ### Author Response · Authors · 2024-11-22
> **Official Comment by Authors (2/2)**
>
> **[W5] Discussion on generalization performance of ours.**
>
> Thank you for pointing this out. To ensure a fair comparison with other methods, we intended to utilize separate datasets for our blur synthesizer, i.e., using the same dataset in the training of the blur synthesis model and deblurring model, addressing any concerns about unfair comparisons arising from the use of additional datasets.
> We would like to clarify that our method can improve model generalization even though we use the same dataset in both trainings. Specifically, if a blur dataset contains $3,000$ blur images and the deblurring model is trained over $1,000$ epochs, it generates approximately $3,000,000$ different synthetic blur images by randomly varying blur intensities, directions, and scenes during the deblurring training. This extensive diversity enhances the generalization ability of the deblurring model, enabling it to handle a wide range of unseen blur scenarios.
> Meanwhile, the existing method (ID-Blau [1]) produces and uses $10,000$ synthetic blur images in advance, as it relies on a diffusion model, which is computationally expensive and challenging to adapt for online data augmentation.
> As suggested by the reviewer, we conducted the cross-dataset evaluations. Please refer to the Common Response $1$ for detailed generalization results.
>
> ----
> **[W6] More comparisons of real-world blurred images.**
>
> To address the reviewer's concerns, we have added more results on real-world blur images captured by us in the revised draft. Please refer to Fig. 11 of the Appendix.
>
> ----
> **References**
>
> [1] Wu et al., Id-blau: Image deblurring by implicit diffusion-based reblurring augmentation, CVPR 2024

---

> > ### Comment · Reviewer_BF8H · 2024-11-25
> > **Official comments by Reviewer**
> >
> > Thank you for the detailed responses! After reviewing the clarifications and peer reviewers' comments, I believe the paper meets the acceptance criteria, albeit with marginal strengths. The overall rating could be slightly increased (like to 7) but has not reached a score of 8 (good paper). Therefore, I will maintain the previous rating.

---

> > > ### Author Response · Authors · 2024-11-25
> > > **Official Comment by Authors**
> > >
> > > Dear Reviewer BF8H,
> > >
> > > We are happy to hear that our paper meets the acceptance criteria. The valuable and constructive feedback has helped us clarify our contribution. Thank you once again for the thoughtful review and the opportunity to refine our work. Please feel free to reach out with any additional questions or suggestions.
> > >
> > >
> > > Best regards,
> > >
> > > Authors

---

> ### Author Response · Authors · 2024-11-25
> **Gentle Reminder**
>
> Dear Reviewer BF8H,
>
> Thank you again for your time and efforts in reviewing our paper.
>
> As the discussion period draws close, we kindly remind you that two days remain for further comments or questions. We would appreciate the opportunity to address any additional concerns you may have before the discussion phase ends.
>
> Thank you very much!
>
> Authors

---

### Author Response · Authors · 2024-11-22
**Common Response**

Dear reviewers and AC,

We express our deepest gratitude for all the constructive feedback from reviewers on our draft.
We also appreciate all positive feedback from the reviewers: beneficial for motion deblurring (by BF8H), well-motivated (by RsRS, iVrH), strong experimental results (by BF8H, RsRS, iVrH), and well-written (by RsRS, iVrH). We sincerely thank all the reviewers for their valuable comments and efforts. We have carefully revised our draft with the following additional discussions and experiments:
- Further clarification (Section 1, Section 3.4)
- Demonstration of GeoSyn's generalization ability (Section D)
- Comprehensive analysis of GeoSyn (Section F)
- Detailed explanation about the projected 3D residual vector (Section A)
- More qualitative results (Section G, Section M)

These updates are temporarily highlighted in $\textcolor{blue}{\text{blue}}$ for the convenience to check.
We hope that our responses and revision address all of the concerns, and we are happy to address any more concerns or questions that the reviewers may have. We will do our best to address all concerns throughout the discussion period.

Here, we collected a common question that multiple reviewers have asked and responded to the question as follows.

Thank you very much,

Authors.

----

**Common Response 1: Demonstration of GeoSyn's generalization ability.**

To demonstrate the generalization ability of our method, we conduct cross-dataset evaluations. First, we train both our blur synthesis and deblurring models on RealBlur (i.e., GeoSyn-R) and test on RealBlur [1], RSBlur [2], and BSD [3]. Additionally, we train our blur synthesis model on GoPro [4] and deblurring model on RealBlur (i.e., GeoSyn-G) and test again on RealBlur, RSBlur and BSD.
As shown in the table below, our GeoSyn-R shows remarkable performance on RealBlur, and GeoSyn-G demonstrates promising generalization performance on RSBlur and BSD. This relies on what dataset is used for training our blur synthesizer. Specifically, the GeoSyn-R is trained on RealBlur, enabling it to generate diverse and dataset-specific blur patterns that contribute to performance improvement on RealBlur. Even though the GeoSyn-R uses only RealBlur in both trainings, it also shows good generalization results on RSBlur and BSD because it can generate numerous and diverse blur patterns during the deblurring training.
On the other hand, our GeoSyn-G leverages separate datasets for the blur synthesizer (GoPro) and the deblurring model (RealBlur), enabling it to benefit from multiple blur datasets.
As a result, it achieves superior generalization performance  (see the results on BSD Test set).


| Train (Deblur) | Test     | Train (Synthesizer) | Methods   | NAFNet | MIMO-UNet+ |
|----------------|----------|---------------------|-----------|---------------|-------------------|
| RealBlur       | RealBlur |          -          | No Aug    | 32.50         | 31.92            |
|                |          |        GoPro        | ID-Blau   | 32.70         | 31.96            |
|                |          |       RealBlur      | GeoSyn-R  | **32.99**         | **32.55**            |
|                |          |        GoPro        | GeoSyn-G  | 32.94         | 32.47                |
| RealBlur       | RSBlur   |          -          | No Aug    | 30.61         | 29.72            |
|                |          |        GoPro        | ID-Blau   | 30.90         | 29.43            |
|                |          |       RealBlur      | GeoSyn-R  | 30.91         | **29.95**            |
|                |          |        GoPro        | GeoSyn-G  | **30.98**             | 29.81                |
| RealBlur       | BSD      |          -          | No Aug    | 29.67         | 29.25            |
|                |          |        GoPro        | ID-Blau   | 30.42         | 28.93            |
|                |          |       RealBlur      | GeoSyn-R  | 30.83         | 29.73            |
|                |          |        GoPro        | GeoSyn-G  | **30.91**             | **29.91**                |

----
**References**

[1] Rim et al., Real-world blur dataset for learning and benchmarking deblurring algorithms, ECCV 2020

[2] Rim et al., Realistic blur synthesis for learning image deblurring, ECCV 2022

[3] Zhong et al., Efficient spatio-temporal recurrent neural network for video deblurring, ECCV 2020

[4] Nah et al., Deep multi-scale convolutional neural network for dynamic scene deblurring, CVPR 2017

---

### Meta-Review · Area_Chair_vfiV · 2024-12-23

**Metareview:**

This paper focuses on augmenting training data for image deblurring algorithms. Because existing datasets have limited scope and variation, a more realistic data synthesis method would be a valuable contribution to the community. Overall, the reviewers find the idea well-motivated and regard the experimental results as comprehensive. Following the rebuttal and author-reviewer exchange, all reviewers agree that this paper meets the acceptance threshold. The AC concurs with the reviewers’ opinions and recommends acceptance of this paper.

**Additional Comments On Reviewer Discussion:**

Reviewers offered very professional and insightful feedback, such as ways to achieve more realistic yet flexible manipulation and a thorough investigation of the method’s generalization capabilities. Although the authors endeavored to address these points, some issues remain unresolved. We hope the community finds inspiration in these discussions and continues to advance solutions in this area.

---

### Decision · Program_Chairs · 2025-01-22

Accept (Poster)